# AnTKV: Anchor Token-Aware Ultra-low-bit Vector Quantization for KV Cache in Large Language Models

## Abstract

Quantization has emerged as an effective and lightweight solution to reduce the memory footprint of the KV cache in Large Language Models. Nevertheless, minimizing the accuracy degradation caused by ultra-low-bit KV cache quantization remains a significant challenge. While scalar quantization is constrained by 1-bit bound, vector quantization exploits intra-vector correlations and enables sub-bit regimes, making it more suitable for ultra-low-bit quantization. To further mitigate quantization-induced degradation, we reveal that the degradation is highly uneven across tokens in attention quality. To investigate this unevenness, we introduce anchor score to measure each token's sensitivity to quantization. Our analysis and experiments show that preserving a small subset (1%) of tokens with the highest Anchor Score significantly mitigates accuracy loss under aggressive quantization. We propose `AnTKV`, a dual-stage framework that leverages anchor token-aware vector quantization to compress the KV cache. It combines offline token-aware centroids learning and online anchor token selection to balance compression and accuracy. To enable efficient deployment, we design an online anchor token selection kernel compatible with FlashAttention. It allows LLaMA3-8B to scale to 840K tokens on a single 80GB A100, while delivering up to $3.5\times$ higher decoding throughput over the FP16 baseline. Experiments demonstrate that `AnTKV` matches or surpasses prior methods at 4-bit, and significantly reduce perplexity under ultra-low-bit quantization, achieving 6.32 at 1-bit on Mistral-7B, compared to 7.25 for CQ and 15.36 for KVQuant.

## 1 Introduction

Large Language Models (LLMs) have gained wide attention owing to their remarkable capabilities in diverse applications OpenAI (2023); Guo et al. (2025); Team et al. (2023); Tang et al. (2025); Dong et al. (2025). With rapid recent advances, LLMs currently handle context lengths from hundreds of thousands to millions of tokens, enabling them to tackle increasingly complex tasks Zhao et al. (2023); Wang et al. (2025b); Das et al. (2025); Wang et al. (2025a). Most LLMs adopt decoder-based transformer architectures, where tokens are generated autoregressively and the KV cache grows rapidly with context length Zhu et al. (2025b); Bai et al. (2024); Liu et al. (2025a) and batch size increases Pope et al. (2022). The large amount of memory footprint of the KV cache poses a significant challenge. For example, with LLaMA-3 at 128K tokens it already approaches the model size, and for million-token models like Gemini it becomes prohibitive. Beyond inference, LLM-RAG systems Yao et al. (2025) pre-generate and store massive KV caches, creating additional storage challenges.

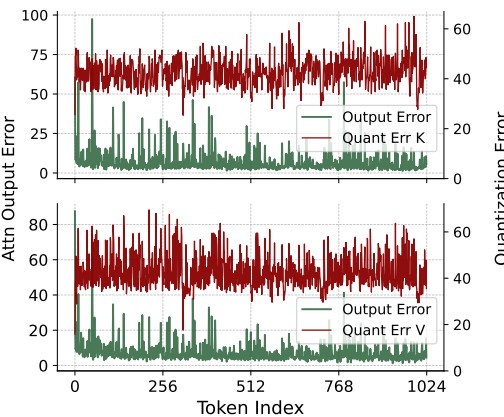 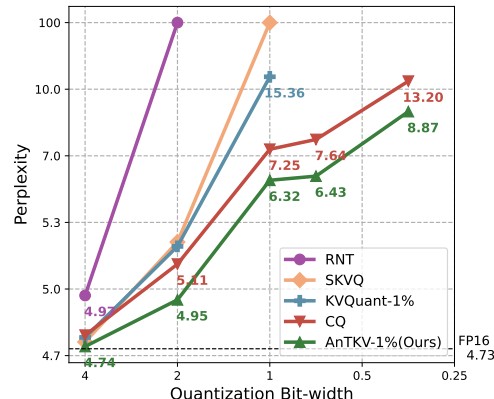

Figure 1: The $L_1$ norm error of attention output when quantizing the $i$th token's KV cache in Mistral-7B to 1-bit.

Figure 2: The Perplexity of Mistral-7B on the WikiText-2 across different quantization bit-widths.

To address the substantial size of KV caches, numerous techniques have been explored Liu et al. (2025c;b; 2024a); Zhang et al. (2024c); Liu et al. (2024b); Dong et al. (2025), with quantization proving to be especially effective. Quantization methods are generally classified into scalar quantization (SQ) and vector quantization (VQ). SQ compresses the KV cache by mapping floating-point values to fixed low-bit representations Liu et al. (2024b); Duanmu et al. (2024); Hooper et al. (2024), but the 1-bit lower bound constrains its maximum compression ratio to 1/16 of FP16. In contrast, VQ compresses high-dimensional vectors by mapping them to a finite codebook Lingle (2024); Zhang et al. (2024b), which captures intra-vector correlations, achieves superior compression ratios and fidelity under ultra-low-bit quantization.

We observe that tokens contribute unequally to model accuracy during inference. To illustrate this, we quantize the KV cache of Mistral-7B Jiang et al. (2023) to 1-bit using VQ. Figure 1 depicts the attention output $L_1$ norm error in the 31st layer when quantizing the KV cache across different token indices, and similar error distributions are observed in other layers. As evident from the figure, although per-token quantization errors are similar, the resulting error in the attention output vary widely across tokens and a small subset exhibits error that are tens of times larger when quantized (anchor tokens). Quantifying token importance is essential for unlocking the full potential of ultra-low-bit KV cache quantization.

To address this, we conduct a forward error analysis Boldo & Melquiond (2017) on attention-based models with respect to KV, and then propose `AnTKV`, a dual-stage framework for KV cache quantization. In the offline stage, `AnTKV` performs token-aware weighted k-means clustering to generate centroids, where the weights are derived from error propagation factors obtained through forward error analysis. Tokens that cause larger increases in output error are assigned larger weights during clustering. However, the propagation factors entail substantial gradient cost. To overcome this limitation, we propose **An**chor **S**core (AnS) derived from the forward error analysis of the attention, which quantifies the output sensitivity to quantizing each token's KV cache. At inference time, AnS is computed for each prompt to identify this small token subset. `AnTKV` handles this subset of tokens in a simple yet effective manner by preserving them in full precision, thereby reducing accuracy loss. Moreover, we design and implement a lightweight GPU kernel for AnS computation. More specifically, we extend FlashAttention Dao (2023) to store low memory overhead softmax intermediate results, thereby enabling efficient online anchor token selection.

We conduct extensive experiments to evaluate the effectiveness of `AnTKV` across various quantization bit-widths on a range of LLMs, including the LLaMA-2/3 Touvron et al. (2023); Grattafiori et al. (2024) and Mistral-7B Jiang et al. (2023) families. As shown in Figure 2, `AnTKV` consistently outperforms existing approaches across bit-widths from 4-bit down to 0.375-bit. On WikiText-2 Merity et al. (2016), `AnTKV` achieves a perplexity of 6.32 at 1-bit, reducing error by 0.93 compared to CQ (7.25) Zhang et al. (2024b), and by a substantial 9.04 compared to KVQuant-1% (15.36) Hooper et al. (2024). Even under the aggressive 0.375-bit setting, `AnTKV` attains a perplexity of 8.87, surpassing CQ (13.20) by 4.33. Thanks to the efficient GPU implementation of AnS, `AnTKV` also

scales to extremely long contexts: on LLaMA3-8B, it supports up to 840K tokens under 0.375-bit quantization on a single A100-80GB GPU. Moreover, during decoding, AnTKV increases the maximum batch size by $3.3\times$ and improves throughput by $3.4\times$ at a 1K context length compared to full precision.

To summarize, we make the following contributions in this work.

- To the best of our knowledge, this work is the first to investigate the feasibility of quantization the KV cache to sub-bit while preserving model accuracy.
- We identify that different tokens contribute unequally to model accuracy under quantization and highlight the existence of anchor tokens that dominate output error.
- We propose AnTKV, which performs token-aware weighted clustering offline and leverages AnS online to efficiently identify anchor tokens, preserving them in full precision to mitigate accuracy loss, and we evaluate it on the LLaMA2/3 and Mistral families, where it consistently improves performance across a wide range of quantization bit-widths.
- We implement custom GPU kernels for the online stage, enabling AnTKV to scale to 840K tokens on a single GPU and deliver significant throughput gains during decoding.

## 2 BACKGROUND

### 2.1 TRANSFORMER AND ATTENTION

The transformer block has become the fundamental architecture of LLMs; it consists of a fully connected feed-forward network and an attention. The attention enables the model to capture connections among tokens within the context. Specifically, it maps a query ($Q$) and a set of KV pairs ($K$ and $V$) to an output, i.e.,

$$\texttt{Attn}(Q, K, V) = \texttt{Softmax}\left(\frac{QK^T}{\sqrt{d}}\right) V \quad \text{with} \quad Q, K, V \in \mathbb{R}^{n \times d},$$

where $\texttt{Softmax}(\cdot)$ is the softmax operator. Since position information is crucial in LLMs, Rotary Position Embedding (RoPE) Su et al. (2023) is a widely used technique for encoding it into query and key vectors Su et al. (2024), which is denoted as $\widetilde{Q}$ and $\widetilde{K}$ in this paper. With RoPE, the attention score $A$ is rewritten as $\texttt{Softmax}\left(\frac{\widetilde{Q}\widetilde{K}^T}{\sqrt{d}}\right)$, and its $L_p$ "entry-wise" matrix norm is defined as $\left(\sum_{i,j} |A_{i,j}|^p\right)^{1/p}$. Other notations used in this work, such as the Kronecker product, the Hadamard product, and the row-by-row vectorization are denoted as $\otimes$, $\odot$, and $\texttt{Vec}(\cdot)$, respectively.

### 2.2 MEMORY CONSTRAINTS IN LLM INFERENCE

During the process of LLM inference, the key and value generated at prefill and each decoding step are stored to avoid redundant computation. The KV cache is repeatedly accessed in subsequent steps to compute attention over the full context seen so far, significantly accelerating autoregressive decoding. In a multi-turn conversation, the KV cache from previous turns can also be reused during the prefill stage to further reduce latency. Nowadays LLMs, like LLaMA3.1-8B Grattafiori et al. (2024) and Gemini 1.5 Pro Team et al. (2023), now handle much longer context lengths, up to 128K and 2 million tokens, respectively. This increase in context length Liu et al. (2025a); Wang et al. (2025a) causes the GPU memory consumed by KV cache to even exceed the model. The rapid growth of KV cache greatly limits the deployment of LLMs with long context. Quantization is an effective approach to compressing the KV cache, substantially reducing memory footprint while preserving essential information.

## 3 METHODOLOGY

VQ is employed as it enables sub-bit compression, which SQ cannot achieve, and in the ultra-low-bit regime ($\leq 2$ bit) it leverages intra-vector correlations to attain significantly better quantization quality compared to SQ.

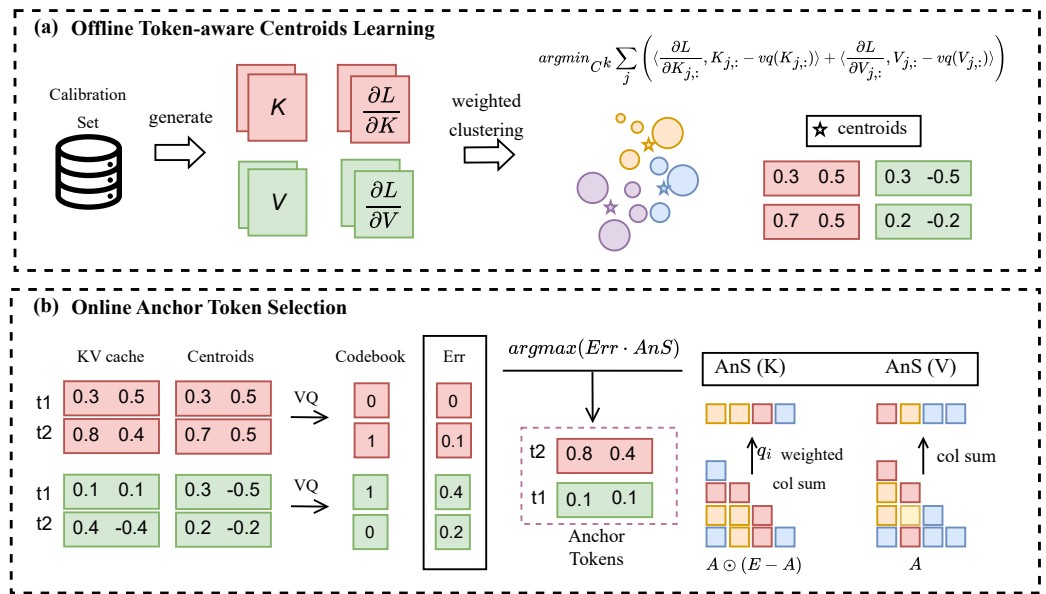

Figure 3: Overview of AnTKV. In the stage (a), token-aware centroids are learned from calibration data through weighted clustering, where the weights are error-propagation factors obtained by forward error analysis. In the stage (b), the KV cache is quantized with centroids, and AnS is computed to identify anchor tokens, which are preserved in full precision to mitigate accuracy loss.

As shown in the figure 1, we observe that although per-token KV cache quantization errors appear similar, the resulting attention output errors differ dramatically. Moreover, the error distribution is steep, with large errors occurring unpredictably across tokens. Previous works on KV cache quantization typically treat all tokens equally or are biased toward attention sinks, as inspired by AttentionSink Xiao et al. (2024b), which we argue is suboptimal. Our observation reveals a new opportunity for KV cache quantization, where tokens with greater influence on attention output error should be prioritized. Building on this observation, we propose AnTKV, a token-aware VQ framework for KV cache quantization. It adopdts a dual-stage design consisting of offline token-aware centroid learning (Figure 3(a)) and online anchor token selection (Figure 3(b)). The central challenge lies in effectively measuring token importance. During centroids leanring, forward error analysis provides per-token error propagation factors, which serve as weights in a token-aware $k$-means clustering to generate VQ centroids, thereby emphasizing tokens with significant influence on attention-output error. For anchor token selection, directly using error propagation factors incurs expensive gradient computation. To avoid this cost, we introduce AnS, a lightweight metric derived from attention forward error analysis that enables efficient prompt-aware identification of anchor tokens.

## 3.1 OFFLINE TOKEN-AWARE CENTROIDS LEARNING

In this stage, centroids are learned offline by aggregating KV cache from calibration data. Specifically, each row (per token) of $\boldsymbol{K}$ and $\boldsymbol{V}$ is divided into several sub-vectors, then using $k$-means to cluster them, and only the centroids are stored. Specifically, each token row of $\boldsymbol{K}$ and $\boldsymbol{V}$ is partitioned into sub-vectors and clustered with $k$-mean. The centroids are retained for online KV cache quantization.

From an accuracy perspective, replacing $\boldsymbol{K}$ and $\boldsymbol{V}$ with their corresponding centroids should introduce minimal impact on model output. Mathematically, it can be formulated as

$$\min_{\boldsymbol{C}^k \in \mathbb{R}^{c^k \times d}} |L(\boldsymbol{K}, \boldsymbol{V}) - L(\mathrm{vq}(\boldsymbol{K}), \mathrm{vq}(\boldsymbol{V}))|, \tag{1}$$

where $c^k$ is the number of clusters, and $\boldsymbol{C}^k$ is the matrix consisting of centroids. By the first-order Taylor series expansion, we have

$$L(\boldsymbol{K}, \boldsymbol{V}) - L(\mathrm{vq}(\boldsymbol{K}), \mathrm{vq}(\boldsymbol{V})) \approx \sum_j \left( \langle \frac{\partial L}{\partial \boldsymbol{K}_{j,:}}, \boldsymbol{K}_{j,:} - \mathrm{vq}(\boldsymbol{K}_{j,:}) \rangle + \langle \frac{\partial L}{\partial \boldsymbol{V}_{j,:}}, \boldsymbol{V}_{j,:} - \mathrm{vq}(\boldsymbol{V}_{j,:}) \rangle \right), \tag{2}$$

where $\frac{\partial L}{\partial \boldsymbol{K}_{j,:}}$ and $\frac{\partial L}{\partial \boldsymbol{V}_{j,:}}$ are the gradient of the loss function $L(\cdot)$ with respect to $\boldsymbol{K}_{j,:}$ and $\boldsymbol{V}_{j,:}$, i.e., the $j$th row of $\boldsymbol{K}$ and $\boldsymbol{V}$ that correspond to the $j$th token. From equation equation 1 and equation 2, the weighted k-means clustering with gradients as weights is essentially to find a clustering strategy that minimizes the error caused by KV quantization.

## 3.2 Online Anchor Token Selection

In online inference, gradients can no longer serve as the token importance metric due to the cost imposed by real-time constraints. To address this, we perform an error propagation analysis of the attention operator (`Attn`). Specifically, the analysis derives a perturbation bound of `Attn` with respect to each row of $\boldsymbol{K}$ and $\boldsymbol{V}$, as presented in Theorem 1, with the detailed proof provided in Appendix C.

**Theorem 1.** *Let $\delta \boldsymbol{K}$ and $\delta \boldsymbol{V}$ be the error perturbation terms corresponding to $\boldsymbol{K}$ and $\boldsymbol{V}$ respectively, and satisfy*

$$\|\delta \boldsymbol{K}\|_{L_1} \ll \|\boldsymbol{K}\|_{L_1} \quad and \quad \|\delta \boldsymbol{V}\|_{L_1} \ll \|\boldsymbol{V}\|_{L_1}.$$

*Then we have*

$$\begin{aligned}
&\|\texttt{Attn}(\boldsymbol{Q}, \boldsymbol{K} + \delta \boldsymbol{K}, \boldsymbol{V}) - \texttt{Attn}(\boldsymbol{Q}, \boldsymbol{K}, \boldsymbol{V})\|_{L_1} \\
&\lesssim \sum_j \sum_i \left\|\left(\boldsymbol{V}^\top \mathrm{Diag}(\boldsymbol{A}_{i,:})(\boldsymbol{I}_n - \boldsymbol{e}\boldsymbol{A}_{i,:})\right)_{:,j}\right\|_{L_1} \|\boldsymbol{Q}_{i,:}\|_{L_2} \|\delta \boldsymbol{K}_{j,:}\|_{L_1}
\end{aligned} \tag{3}$$

*and*

$$\|\texttt{Attn}(\boldsymbol{Q}, \boldsymbol{K}, \boldsymbol{V}) - \texttt{Attn}(\boldsymbol{Q}, \boldsymbol{K}, \boldsymbol{V} + \delta \boldsymbol{V})\|_{L_1} \leq \sum_j \|\boldsymbol{A}_{:,j}\|_{L_1} \|\delta \boldsymbol{V}_{j,:}\|_{L_1}, \tag{4}$$

*where $\boldsymbol{e} \in \mathbb{R}^n$ is a vector whose entries are all 1.*

We remark that the error propagation factors corresponding to $\boldsymbol{K}_{j,:}$ and $\boldsymbol{V}_{j,:}$ given in Theorem 1 can be regarded as the upper bound of the gradient of the attention operator related to $\boldsymbol{K}_{j,:}$ and $\boldsymbol{V}_{j,:}$.

The computation involving $\boldsymbol{K}$ in Theorem 1 introduces significant overhead and is therefore unsuitable for online inference. To address this limitation, we propose a simplified variant that excludes the contribution of $\boldsymbol{V}$ to the quantization error of $\boldsymbol{K}$. This leads to the following reformulation, with a detailed evaluation of AnS effectiveness provided in Appendix D.

$$\text{AnS}(\boldsymbol{V}_{j,:}) = \sum_i \boldsymbol{A}_{i,j} \qquad \text{AnS}(\boldsymbol{K}_{j,:}) = \sum_i \boldsymbol{A}_{i,j}(1 - \boldsymbol{A}_{i,j}) \cdot \|\boldsymbol{Q}_{i,:}\|_2 \tag{5}$$

In online inference, during the prefill phase, AnS serves as an effective metric for identifying anchor tokens that induce substantial accuracy loss. In autoregressive decoding phase, AnS can still be computed. However, the anchor tokens it identifies may already have been quantized, which prevents preserving their full-precision values and limits error reduction. An important observation is that both AnS(K) and AnS(V) exhibit strong locality during the decoding phase (see Appendix E), with anchor tokens predominantly concentrated at the head and tail of the sequence. Experimental results show that the anchor tokens at the head of the sequence are consistently identified during the prefill stage, corresponding to sink tokens. This observation further demonstrates the effectiveness of our method. Building on the tail locality, we use a sliding-window approximation of AnS during decoding to further enhance efficiency while mitigating accuracy degradation.

## 3.3 Implementation

For the offline centroids learning stage, the gradients of $\boldsymbol{K}$ and $\boldsymbol{V}$ are employed as weights for centroid learning. We implement it with a custom LinearWithAct to capture KV cache and corresponding gradients. Subsequently, we employ the weighted k-means provided by cuML to perform efficient clustering.

In the online stage, AnS is derived from the error propagation factor given in Theorem 1 and Equation equation 5. To enable efficient long-context inference, we design and implement a dedicated GPU kernel using Triton that computes AnS in conjunction with FlashAttention. Because AnS

requires reduction operations over the attention score matrix $A$ and its transformed form $A \odot (E - A)$ along the query dimension (column-wise), direct fusion into FlashAttention is infeasible. To preserve the efficiency of FlashAttention, we decouple AnS computation and execute it immediately afterward. For this purpose, we extend FlashAttention to additionally output three auxiliary tensors: the $L_2$ norm of each query vector, the key-wise (row-wise) sum, and the key-wise maximum of the matrix $QK^T$. These tensors allow the reconstruction of the attention scores and facilitate AnS computation with minimal overhead. Further implementation details are provided in Algorithm 1 of Appendix F.

Finally, since the application of RoPE disrupts the channel-wise magnitude distribution of $K$ (see Appendix B), which otherwise exhibits large inter-cluster distances and small intra-cluster variances, the pre-RoPE strategy, consistent with Hooper et al. (2024), is adopted in `AnTKV`.

## 4 EXPERIMENTS

In this section, we present an extensive comparison between `AnTKV` and existing KV quantization methods. The experimental setup is detailed below.

**Models, Datasets, Metrics, and Parameter Settings.** To validate the effectiveness and generality of `AnTKV` in KV cache quantization, we evaluate five representative models from the LLaMA and Mistral families. For calibration, 128 samples of length 2048 are drawn from the WikiText2 training set. Model quality is assessed through three categories of benchmarks: (i) perplexity on WikiText-2 and C4; (ii) zero-shot accuracy on MMLU Hendrycks et al. (2021), ARC-C Clark et al. (2018), MathQA Amini et al. (2019), and PIQA Bisk et al. (2020) to evaluate understanding and reasoning; and (iii) long-context performance on LongBench Bai et al. (2024). For perplexity and zero-shot evaluations, quantized KV caches are directly used for attention outputs, whereas for LongBench, full precision KV cache is used to compute attention outputs and quantized KV cache is used during decoding. Across all benchmarks, anchor tokens are restricted to a small subset: 1% of the context length for perplexity, 16 for understanding and reasoning, and 64 for LongBench. For fair comparison, a sliding window of size 32 is applied in LongBench, following the mainstream setting.

**Baselines.** We compare `AnTKV` with full precision and representative KV cache quantization methods, including KIVI Liu et al. (2024b), SKVQ Duanmu et al. (2024), KVQuant-1% Hooper et al. (2024), and CQ Zhang et al. (2024b). SKVQ is configured with a group size of 64 and five sink tokens Xiao et al. (2024b), while KVQuant retains four sink tokens. Since CQ results are not publicly available, we reproduced them following the methodology in their paper to the best of our understanding. For VQ settings, we adopt the notation "d$n$c$m$", covering 4-bit (d2m256), 2-bit (d4m256), 1-bit (d8m256), 0.75-bit (d16m4096), and 0.375-bit (d32m4096).

### 4.1 PERPLEXITY RESULTS

Perplexity is a standard benchmark that is widely used to evaluate the quality of the output of LLMs, with lower values indicating better performance. The perplexity results for different KV quantization approaches on WikiText-2 and C4 are presented in Table 1. The results in this table indicate that the proposed `AnTKV` consistently achieves competitive or superior perplexity across various bit-widths and model architectures. Under 4-bit and 2-bit quantization, it achieves competitive performance compared to baseline. In the 1-bit and sub-bit regimes, it significantly outperforms all baselines. On the C4 dataset under sub-bit quantization, baseline methods suffer from extremely high perplexity, as fixed centroids fail to capture anchor tokens. By contrast, with its effective AnS design and anchor token selection, `AnTKV` substantially lowers perplexity, reducing it from 66.28 to 14.42 on LLaMA-3-8B at 0.75-bit.

### 4.2 UNDERSTANDING AND REASONING BENCHMARK

To assess the breadth of `AnTKV`'s understanding and reasoning capabilities, we evaluate it on four representative benchmarks using LLaMA-8B-Instruct and Mistral-7B-Instruct. These benchmarks target multi-domain knowledge reasoning (MMLU), complex question answering (ARC-Challenge), commonsense reasoning (PIQA), and mathematical problem solving (MathQA). Due to missing the implementations in the official repository, KIVI and KVQuant are not included. As

Table 1: All evaluations are performed under the maximum context length of each model, specifically 4096 for LLaMA-2-7B and 8192 for LLaMA-3-8B and Mistral-7B. "Ours" refers to `AnTKV` without anchor tokens, whereas "Ours-1%" denotes `AnTKV` with 1% of tokens designated as anchor tokens and retained in FP16. For clarity, the reported bit-widths exclude the contribution of centroids.

| | Bit | LLaMA-2-7B | | LLaMA-3-8B | | Mistral-7B | |
|---|---|---|---|---|---|---|---|
| Dataset | | WikiText2 | C4 | WikiText2 | C4 | WikiText2 | C4 |
| Baseline | 16 | 5.12 | 6.63 | 5.54 | 7.10 | 4.73 | 5.66 |
| RTN | | 5.66 | 7.31 | 7.89 | 8.79 | 7.34 | 5.91 |
| SKVQ | | 5.16 | 6.67 | 5.64 | 7.19 | 4.97 | 5.68 |
| KVQuant-1% | 4 | **5.13** | **6.65** | **5.56** | **7.12** | 4.78 | 5.72 |
| CQ | | 5.14 | 6.67 | 5.58 | 7.84 | 4.79 | 5.74 |
| Ours | | 5.18 | 6.76 | 5.61 | 7.69 | 4.76 | 5.69 |
| Ours-1% | | 5.15 | 6.68 | 5.59 | 7.16 | **4.74** | **5.67** |
| RTN | | 4708 | 4708 | 2841 | 2113 | 573 | 477 |
| SKVQ | | 5.54 | 7.21 | 6.73 | 8.31 | 5.21 | 6.14 |
| KVQuant-1% | 2 | 5.49 | **7.02** | 6.11 | **7.65** | 5.19 | 6.10 |
| CQ | | 5.42 | 7.23 | 6.09 | 18.71 | 5.11 | 6.17 |
| Ours | | 5.51 | 7.45 | 6.10 | 16.96 | 5.08 | 6.18 |
| Ours-1% | | **5.34** | **7.02** | **5.97** | 7.68 | **4.95** | **5.97** |
| SKVQ | | 12643 | 12819 | 108879 | 86426 | 3524 | 2741 |
| KVQuant-1% | | 21.55 | 51.84 | 14.80 | 13.95 | 15.36 | 14.24 |
| CQ | 1 | 7.75 | 12.49 | 9.56 | 81.74 | 7.25 | 9.89 |
| Ours | | 7.92 | 13.01 | 9.62 | 74.47 | 7.32 | 10.51 |
| Ours-1% | | **6.50** | **9.40** | **8.51** | **12.51** | **6.32** | **8.44** |
| CQ | | 8.39 | 14.32 | 11.18 | 72.05 | 7.64 | 11.72 |
| Ours | 0.75 | 8.21 | 14.27 | 10.41 | 66.28 | 7.41 | 11.72 |
| Ours-1% | | **6.55** | **9.75** | **8.97** | **14.42** | **6.43** | **9.08** |
| CQ | | 14.82 | 33.59 | 22.80 | 103.5 | 13.20 | 26.34 |
| Ours | 0.375 | 13.37 | 30.51 | 17.70 | 103.5 | 11.65 | 23.98 |
| Ours-1% | | **8.75** | **15.86** | **13.41** | **34.08** | **8.87** | **14.87** |

shown, `AnTKV` consistently maintains higher accuracy across LLaMA and Mistral models, with particularly strong advantages at 1-bit and sub-bit settings where baseline methods degrade sharply.

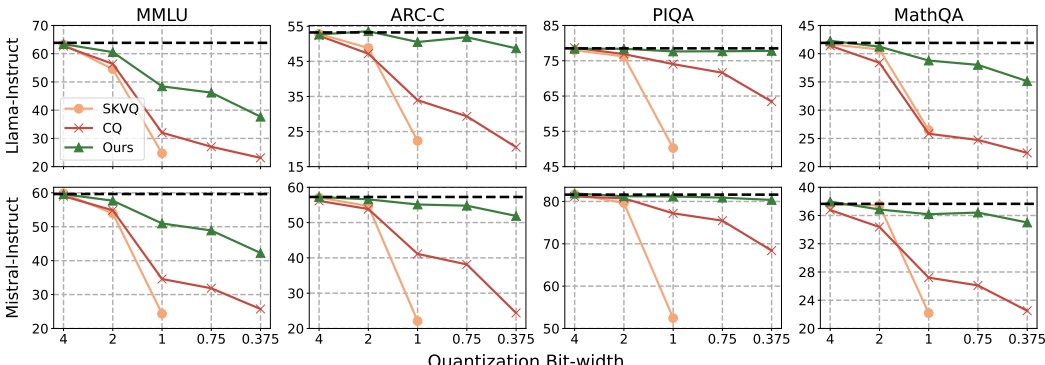

Figure 4: Evaluation of understanding and reasoning accuracy on MMLU, ARC-C, PIQA, and MathQA under different quantization bit-widths.

### 4.3 LONG-CONTEXT BENCHMARK

To validate the effectiveness of `AnTKV` in handling long-context, We conduct several experiments on the LLaMA-8B-Instruct model using the LongBench benchmark, a diverse collection of tasks such as question answering, retrieval, and summarization, designed to systematically evaluate long-context understanding in language models. We report results on eleven representative sub-tasks from LongBench, along with averaged performance. Due to alignment issues of KIVI and SKVQ, we exclude the triviaqa and gov_report sub-tasks from the comparison. As shown in Figure 5, AnTKV preserves nearly FP16 at 4- and 2-bit quantization across almost all tasks. At the 1-bit quantization, the performance of KIVI and SKVQ has a significant drop. In contrast, AnTKV and CQ still maintain a relatively high accuracy. To further investigate the robustness under aggressive compression, we compare AnTKV and CQ in both sub-bit levels. Figure 5 shows that AnTKV consistently outperforms CQ. Notably, despite aggressive quantization down to 0.375-bit, `AnTKV` maintains tolerable degradation, with the average score decreasing from 46.5 to 38.1.

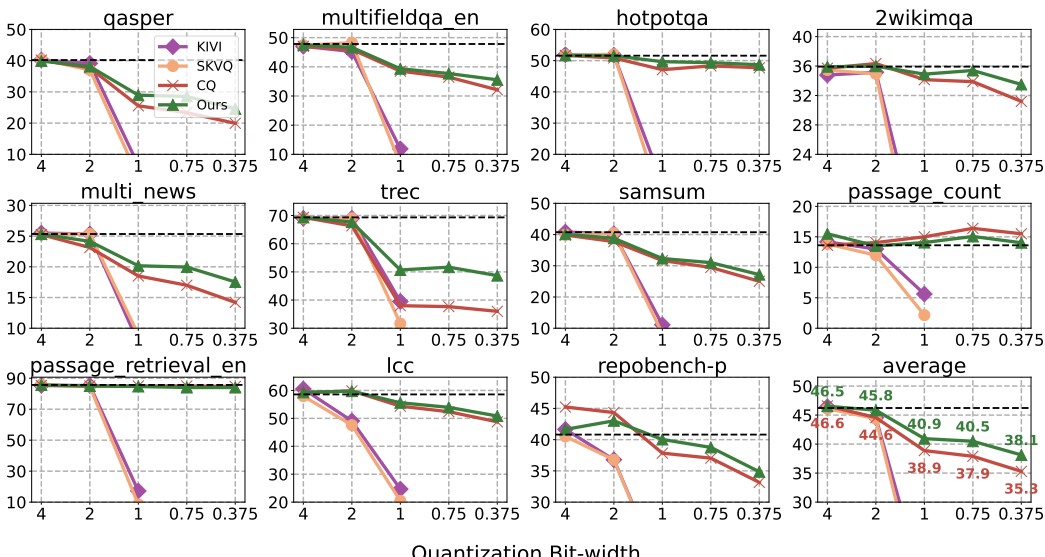

Figure 5: The evaluation accuracy results on LongBench under different KV cache quantization bit-widths. `AnTKV` achieves the best average performance under ultra-low-bit quantization.

### 4.4 EFFICIENCY

In this experiment, we evaluate the efficiency of our `AnTKV` implementation compared with hug-gingface baseline Wolf et al. (2020) on LLaMA-3-8B using a single A100-80GB GPU. As shown in Figure 6, `AnTKV` substantially extends the maximum context length from 128K to 384K. In long-context inference, our profiling shows that intermediate activations account for a substantial portion of memory usage. By introducing a series of in-place operators, `AnTKV` supports up to 810K tokens under 1-bit quantization and 840K under 0.375-bit quantization, while maintaining low memory consumption. To evaluate decoding efficiency, we measure the throughput of `AnTKV` with a fixed context length of 1K tokens. As shown in Figure 7, `AnTKV` enables substantially larger batch sizes and improves throughput across all bit-widths by reducing KV cache access. In particular, under 1-bit quantization, the maximum throughput reaches $3.5\times$.

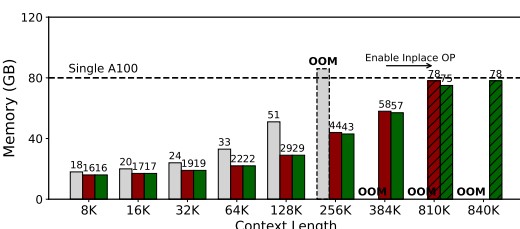

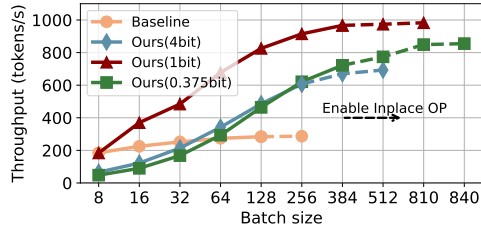

Figure 6: KV cache memory size comparison. Gray bars denote full precision, red bars 1-bit, and green bars 0.375-bit quantization. Striped bars indicate results with in-place operators enabled.

Figure 7: Decoding throughput comparison. Our method supports larger batch sizes and achieves higher throughput. Dashed lines indicate results with in-place operators enabled.

## 4.5 ABLATION STUDY

We conduct a series of experiments to answer the following questions.

**Q1: How does the model performance change as the number of anchor tokens increases?**

As the number of anchor tokens increases, the performance loss decreases rapidly at first, as shown in Table 1. However, the marginal benefit diminishes with more anchor tokens. Detailed results are provided in Appendix G.

**Q2: Does the calibration set affect the performance?**

We find that for VQ-based methods, the calibration set does have some impact on performance under low-bit settings. However, as shown in Table 1 (LLaMA-3-8B, 1-bit, C4), retaining anchor tokens effectively mitigates the performance drop caused by calibration set variation. More detailed results can be found in Appendix H.

## 5 LIMITATION & CONCLUSION

Although `AnTKV` demonstrates its advantages in experiments, it also has few limitations. First, more accurate AnS for tokens and higher performance implementations for its computation may be possible. AnTKV demonstrates strong potential in LLM serving by substantially reducing the size of the KV cache, which in turn alleviates I/O and memory constraints to a significant extent. Nevertheless, further empirical validation is required.

This work addresses the preservation of accuracy under ultra low bit KV cache quantization. We propose `AnTKV` , a vector quantization based framework that exploits intra vector correlations. AnTKV uses a dual stage design with offline token aware centroid learning and online anchor token selection, which mitigates the disproportionate error from anchor tokens. Across the LLaMA and Mistral families, `AnTKV` attains accuracy close to full precision and consistently surpasses baselines in the ultra low bit regime. It also scales LLaMA-3-8B to 840K tokens on a single 80 GB A100, and increases decoding throughput by up to $3.5\times$.

## REPRODUCIBILITY STATEMENT

Our implementation builds on the Hugging Face Transformers library Wolf et al. (2020). The Anchor Score computation as well as the vector quantization and dequantization operators are implemented in Triton for efficiency. We will release the full source code upon acceptance of the paper to ensure reproducibility.

## ETHICS STATEMENT

All experiments in this work are conducted using publicly available models and datasets. We strictly follow the corresponding licenses.

### MODELS

Here, we list all of the model checkpoints used in our experiments:

- LLaMA-2-7B `https://huggingface.co/meta-llama/Llama-2-7b`
- LLaMA-3-8B `https://huggingface.co/meta-llama/Meta-Llama-3-8B`
- LLaMA-3-8B-Instruct `https://huggingface.co/meta-llama/Meta-Llama-3-8B-Instruct`
- Mistral-7B `https://huggingface.co/mistralai/Mistral-7B-v0.1`
- Mistral-7B-Instruct `https://huggingface.co/mistralai/Mistral-7B-Instruct-v0.3`

### DATASETS

We use the following publicly available datasets:

- WikiText2 `https://huggingface.co/datasets/mindchain/wikitext2`
- C4 `https://huggingface.co/datasets/allenai/c4`
- MMLU `https://huggingface.co/datasets/cais/mmlu`
- ARC-C `https://huggingface.co/datasets/allenai/ai2_arc`
- PIQA `https://huggingface.co/datasets/ybisk/piqa`
- MathQA `https://huggingface.co/datasets/allenai/math_qa`
- LongBench `https://huggingface.co/datasets/THUDM/LongBench`

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

# A  RELATED WORKS

**KV cache quantization** A variety of KV cache quantization methods have been proposed to address the memory bottleneck in long-context LLMs Liu et al. (2024b); Duanmu et al. (2024); Hooper et al. (2024); Zhang et al. (2024b); Zhu et al. (2025a). KIVI Liu et al. (2024b) mitigates quantization error by applying per-channel key quantization and employing a sliding window to emphasize locally relevant tokens. SKVQ Duanmu et al. (2024) further explores this direction by introducing channel reordering and clipping. To further reduce accuracy loss, KVQuant Hooper et al. (2024) introduces pre-RoPE key quantization, non-uniform format and element-wise outlier. CQ Zhang et al. (2024b) adopts a VQ-based approach, aiming to exploit cross-channel correlations to further compress the KV cache.

**KV cache compression** Beyond quantization, the field of LLMs is actively exploring advanced methods for KV cache compression. Sparse attention aims to reduce memory footprint by selectively handling the KV cache in a token-wise manner Xiao et al. (2024b); Chen et al. (2024a); Zhu et al. (2025b); Liu et al. (2025c;b); Li et al. (2024). However, it discards the KV cache of a subset of tokens, even though the corresponding tokens may be required in subsequent decoding. Token Merging reduces memory usage by consolidating the KV caches of similar tokens during inference, achieving an effect related to sparse attention but through merging rather than dropping tokens Zhang et al. (2024c); Wang et al. (2024). Retrieval-based methods Liu et al. (2024a); Chen et al. (2024b); Zhang et al. (2024a) offload and index KV caches, retrieving a subset of relevant entries for each query, but introduce additional communication overhead.

**Model Compression** Numerous model compression techniques share common objectives and methodological foundations with KV cache compression. GPTQ Frantar et al. (2023) utilizes calibration set to reduce quantization induced degradation, while SmoothQuant Xiao et al. (2024a) and AWQ Lin et al. (2024) minimize output error from the perspective of error propagation analysis. VQ-based methods such as QUIP# Tseng et al. (2024) further enhance compression fidelity through Hadamard transform. Pruner-Zero Dong et al. (2024b) and Parzc Dong et al. (2024a), explore how to sparsify model weights while preserving model performance. System-level works like Atom Zhao et al. (2024) and QServe Lin et al. (2025) Recent efforts jointly quantize model, KV cache and activatioin, enabling inference under low-bit and leveraging low-precision Tensor Cores to improve system performance, while approaches such as FlashLLM Xia et al. (2023) and Spinfer Fan et al. (2025) accelerate inference by leveraging model sparsity.

# B  DISTRIBUTION OF PRE- AND POST- ROPE KEY

To identify a quantization strategy better suited for $K$ vertor quantization, we compare the distribution of $K$ before and after applying RoPE. Figure 8 presents a visualization of the pre- and post- RoPE $K$. We observe that, compared to the post-RoPE $K$, the pre-RoPE $K$ exhibits smaller inter-cluster distances and lower intra-cluster variance, which contributes to reduced quantization error.

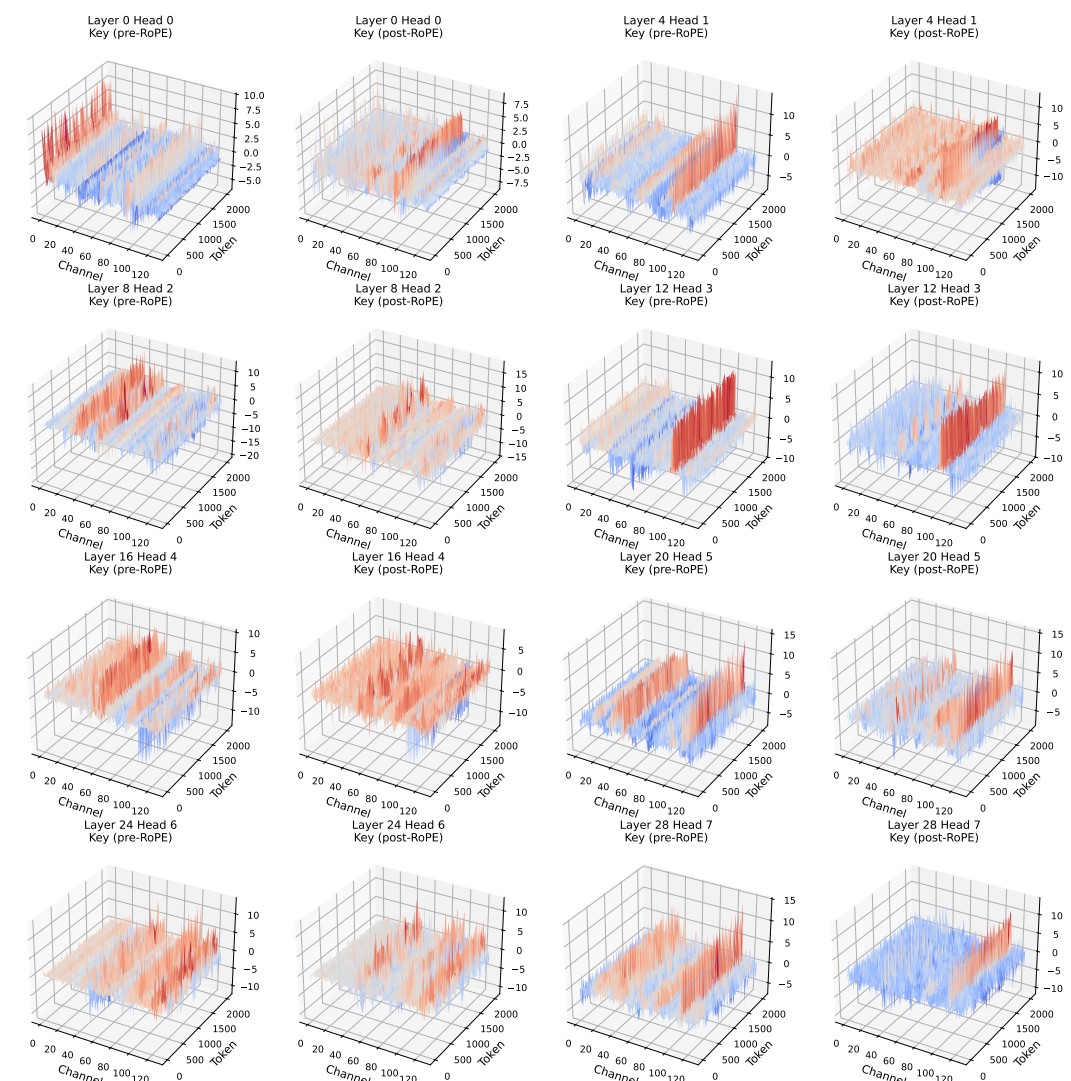

Figure 8: Distribution of pre- and post- RoPE Key. We sampled a 2048-length sentence from WikiText2 and generated pre- and post- RoPE Key on the LLaMA-3-8B model.

## C  PROOF OF THEOREM 1

For $\boldsymbol{K}$, we have

$$\|\texttt{Attn}\left(\boldsymbol{Q},\boldsymbol{K}+\delta\boldsymbol{K},\boldsymbol{V}\right)-\texttt{Attn}\left(\boldsymbol{Q},\boldsymbol{K},\boldsymbol{V}\right)\|_{L_1} = \|\left(\texttt{Softmax}\left(\frac{\widetilde{\boldsymbol{Q}}\widetilde{\boldsymbol{K}}^T}{\sqrt{d}} + \frac{\widetilde{\boldsymbol{Q}}\widetilde{\delta\boldsymbol{K}}^T}{\sqrt{d}}\right) - \texttt{Softmax}\left(\frac{\widetilde{\boldsymbol{Q}}^T\widetilde{\boldsymbol{K}}}{\sqrt{d}}\right)\right)\boldsymbol{V}\|_{L_1}.$$

(6)

The key to estimating the bound of equation 6 lies in the analysis of

$$\texttt{Softmax}\left(\frac{\widetilde{\boldsymbol{Q}}\widetilde{\boldsymbol{K}}^T}{\sqrt{d}} + \frac{\widetilde{\boldsymbol{Q}}\widetilde{\delta\boldsymbol{K}}^T}{\sqrt{d}}\right) - \texttt{Softmax}\left(\frac{\widetilde{\boldsymbol{Q}}\widetilde{\boldsymbol{K}}^T}{\sqrt{d}}\right),$$

(7)

whose $(i,j)$th entry is represented as

$$\frac{\exp\left(\frac{\widetilde{\boldsymbol{Q}}_{i,:}\widetilde{\boldsymbol{K}}_{j,:}^T}{\sqrt{d}} + \frac{\widetilde{\boldsymbol{Q}}_{i,:}\widetilde{\delta\boldsymbol{K}}_{j,:}^T}{\sqrt{d}}\right)}{\sum\limits_{s}\exp\left(\frac{\widetilde{\boldsymbol{Q}}_{i,:}\widetilde{\boldsymbol{K}}_{s,:}^T}{\sqrt{d}} + \frac{\widetilde{\boldsymbol{Q}}_{i,:}\widetilde{\delta\boldsymbol{K}}_{s,:}^T}{\sqrt{d}}\right)} - \frac{\exp\left(\frac{\widetilde{\boldsymbol{Q}}_{i,:}\widetilde{\boldsymbol{K}}_{j,:}^T}{\sqrt{d}}\right)}{\sum\limits_{s}\exp\left(\frac{\widetilde{\boldsymbol{Q}}_{i,:}\widetilde{\boldsymbol{K}}_{s,:}^T}{\sqrt{d}}\right)}.$$

(8)

Since $\|\delta\boldsymbol{K}\|_{L_1} \ll \|\boldsymbol{K}\|_{L_1}$, and by the first-order approximation $\exp(x+\delta x) \approx \exp(x)(1+\delta x)$, equation 8 can be approximated as

$$\frac{\exp\left(\frac{\widetilde{\boldsymbol{Q}}_{i,:}\widetilde{\boldsymbol{K}}_{j,:}^T}{\sqrt{d}}\right)\left(1+\frac{\widetilde{\boldsymbol{Q}}_{i,:}\widetilde{\delta\boldsymbol{K}}_{j,:}^T}{\sqrt{d}}\right)\left(\sum_s \exp\left(\frac{\widetilde{\boldsymbol{Q}}_{i,:}\widetilde{\boldsymbol{K}}_{s,:}^T}{\sqrt{d}}\right)\right)-\exp\left(\frac{\widetilde{\boldsymbol{Q}}_{i,:}\widetilde{\boldsymbol{K}}_{j,:}^T}{\sqrt{d}}\right)\left(\sum_s \exp\left(\frac{\widetilde{\boldsymbol{Q}}_{i,:}\widetilde{\boldsymbol{K}}_{s,:}^T}{\sqrt{d}}\right)\left(1+\frac{\widetilde{\boldsymbol{Q}}_{i,:}\widetilde{\delta\boldsymbol{K}}_{s,:}^T}{\sqrt{d}}\right)\right)}{\left(\sum_s \exp\left(\frac{\widetilde{\boldsymbol{Q}}_{i,:}\widetilde{\boldsymbol{K}}_{s,:}^T}{\sqrt{d}}\right)\right)\left(\sum_s \exp\left(\frac{\widetilde{\boldsymbol{Q}}_{i,:}\widetilde{\boldsymbol{K}}_{s,:}^T}{\sqrt{d}}\right)\left(1+\frac{\widetilde{\boldsymbol{Q}}_{i,:}\widetilde{\delta\boldsymbol{K}}_{s,:}^T}{\sqrt{d}}\right)\right)}$$

$$=\frac{\exp\left(\frac{\widetilde{\boldsymbol{Q}}_{i,:}\widetilde{\boldsymbol{K}}_{j,:}^T}{\sqrt{d}}\right)}{\sum_s \exp\left(\frac{\widetilde{\boldsymbol{Q}}_{i,:}\widetilde{\boldsymbol{K}}_{s,:}^T}{\sqrt{d}}\right)} \cdot \frac{\sum_s \exp\left(\frac{\widetilde{\boldsymbol{Q}}_{i,:}\widetilde{\boldsymbol{K}}_{s,:}^T}{\sqrt{d}}\right)\left(\frac{\widetilde{\boldsymbol{Q}}_{i,:}\left(\delta\boldsymbol{K}_{j,:}^T-\delta\boldsymbol{K}_{s,:}^T\right)}{\sqrt{d}}\right)}{\sum_s \exp\left(\frac{\widetilde{\boldsymbol{Q}}_{i,:}\widetilde{\boldsymbol{K}}_{s,:}^T}{\sqrt{d}}\right)\left(1+\frac{\widetilde{\boldsymbol{Q}}_{i,:}\widetilde{\delta\boldsymbol{K}}_{s,:}^T}{\sqrt{d}}\right)}$$

$$\approx\frac{\exp\left(\frac{\widetilde{\boldsymbol{Q}}_{i,:}\widetilde{\boldsymbol{K}}_{j,:}^T}{\sqrt{d}}\right)}{\sum_s \exp\left(\frac{\widetilde{\boldsymbol{Q}}_{i,:}\widetilde{\boldsymbol{K}}_{s,:}^T}{\sqrt{d}}\right)} \cdot \frac{\sum_s \exp\left(\frac{\widetilde{\boldsymbol{Q}}_{i,:}\widetilde{\boldsymbol{K}}_{s,:}^T}{\sqrt{d}}\right)\left(\frac{\widetilde{\boldsymbol{Q}}_{i,:}\left(\delta\boldsymbol{K}_{j,:}^T-\delta\boldsymbol{K}_{s,:}^T\right)}{\sqrt{d}}\right)}{\sum_s \exp\left(\frac{\widetilde{\boldsymbol{Q}}_{i,:}\widetilde{\boldsymbol{K}}_{s,:}^T}{\sqrt{d}}\right)}$$

$$=\boldsymbol{A}_{i,j} \cdot \frac{\sum_s \exp\left(\frac{\widetilde{\boldsymbol{Q}}_{i,:}\widetilde{\boldsymbol{K}}_{s,:}^T}{\sqrt{d}}\right)\left(\frac{\widetilde{\boldsymbol{Q}}_{i,:}\left(\delta\boldsymbol{K}_{j,:}^T-\delta\boldsymbol{K}_{s,:}^T\right)}{\sqrt{d}}\right)}{\sum_s \exp\left(\frac{\widetilde{\boldsymbol{Q}}_{i,:}\widetilde{\boldsymbol{K}}_{s,:}^T}{\sqrt{d}}\right)}.$$

For convenience, we denote $\frac{\widetilde{\boldsymbol{Q}}\widetilde{\delta\boldsymbol{K}}^T}{\sqrt{d}}$ and $\left[\frac{\sum_s \exp\left(\frac{\widetilde{\boldsymbol{Q}}_{i,:}\widetilde{\boldsymbol{K}}_{s,:}^T}{\sqrt{d}}\right)\left(\frac{\widetilde{\boldsymbol{Q}}_{i,:}\left(\delta\boldsymbol{K}_{j,:}^T-\delta\boldsymbol{K}_{s,:}^T\right)}{\sqrt{d}}\right)}{\sum_s \exp\left(\frac{\widetilde{\boldsymbol{Q}}_{i,:}\widetilde{\boldsymbol{K}}_{s,:}^T}{\sqrt{d}}\right)}\right]_{n\times n}$ as $\boldsymbol{X}$ and $\boldsymbol{Y}$

respectively. Then equation 7 can be approximated as $(\boldsymbol{A}\odot\boldsymbol{Y})\boldsymbol{V}$, and by the property of Kronecker product, we have

$$\text{Vec}\left((\boldsymbol{A}\odot\boldsymbol{Y})\boldsymbol{V}\right)=\left(\boldsymbol{I}_n\otimes\boldsymbol{V}^T\right)\text{Vec}\left(\boldsymbol{A}\odot\boldsymbol{Y}\right)=\left(\boldsymbol{I}_n\otimes\boldsymbol{V}^T\right)\text{Diag}\left(\text{Vec}(\boldsymbol{A})\right)\text{Vec}(\boldsymbol{Y}).$$

Further, we can obtain

$$\|\text{Attn}\left(\boldsymbol{Q},\boldsymbol{K}+\delta\boldsymbol{K},\boldsymbol{V}\right)-\text{Attn}\left(\boldsymbol{Q},\boldsymbol{K},\boldsymbol{V}\right)\|_{L_1} \approx \|\left(\boldsymbol{I}_n\otimes\boldsymbol{V}^T\right)\text{Diag}\left(\text{Vec}(\boldsymbol{A})\right)\text{Vec}(\boldsymbol{Y})\|_{L_1}$$

$$=\sum_i \|\boldsymbol{V}^T\text{Diag}\left(\boldsymbol{A}_{i,:}\right)\left(\boldsymbol{I}_n-\boldsymbol{e}\boldsymbol{A}_{i,:}\right)\boldsymbol{X}_{i,:}^T\|_{L_1}$$

$$\leq \sum_i\sum_j \|\left(\boldsymbol{V}^T\text{Diag}\left(\boldsymbol{A}_{i,:}\right)\left(\boldsymbol{I}_n-\boldsymbol{e}\boldsymbol{A}_{i,:}\right)\right)_{:,j}\|_{L_1}|\widetilde{\boldsymbol{Q}}_{i,:}\widetilde{\delta\boldsymbol{K}}_{j,:}^T|$$

$$\leq \sum_j\sum_i \|\left(\boldsymbol{V}^T\text{Diag}\left(\boldsymbol{A}_{i,:}\right)\left(\boldsymbol{I}_n-\boldsymbol{e}\boldsymbol{A}_{i,:}\right)\right)_{:,j}\|_{L_1}\|\boldsymbol{Q}_{i,:}\|_2\|\delta\boldsymbol{K}_{j,:}\|_{L_1}.$$

$$(9)$$

For $\boldsymbol{V}$, we have

$$\|\text{Attn}\left(\boldsymbol{Q},\boldsymbol{K},\boldsymbol{V}+\delta\boldsymbol{V}\right)-\text{Attn}\left(\boldsymbol{Q},\boldsymbol{K},\boldsymbol{V}\right)\|_{L_1}=\|\text{Softmax}\left(\frac{\widetilde{\boldsymbol{Q}}\widetilde{\boldsymbol{K}}^T}{\sqrt{d}}\right)\delta\boldsymbol{V}\|_{L_1}$$

$$=\|\boldsymbol{A}\delta\boldsymbol{V}\|_{L_1}=\sum_{i,k}|\sum_j \boldsymbol{A}_{i,j}\delta\boldsymbol{V}_{j,k}|$$

$$\leq \sum_{i,k}\sum_j \boldsymbol{A}_{i,j}|\delta\boldsymbol{V}_{j,k}| \qquad (10)$$

$$=\sum_j \left(\sum_i \boldsymbol{A}_{i,j}\right)\left(\sum_k |\delta\boldsymbol{V}_{j,k}|\right)$$

$$=\sum_j \|\boldsymbol{A}_{:,j}\|_{L_1}\|\delta\boldsymbol{V}_{j,:}\|_{L_1}.$$

## D  THE EFFECTIVENESS OF ANS

We quantized each token across layers and heads, and separately recorded the errors in the attention outputs. As shown in Figure 9 and equation 5, it can be observed that the relative values derived from

our AnS, as defined in Theorem 1, closely align with the actual outputs of the attention. For a fair comparison, the output errors for $K$ also exclude the contribution of $V$.

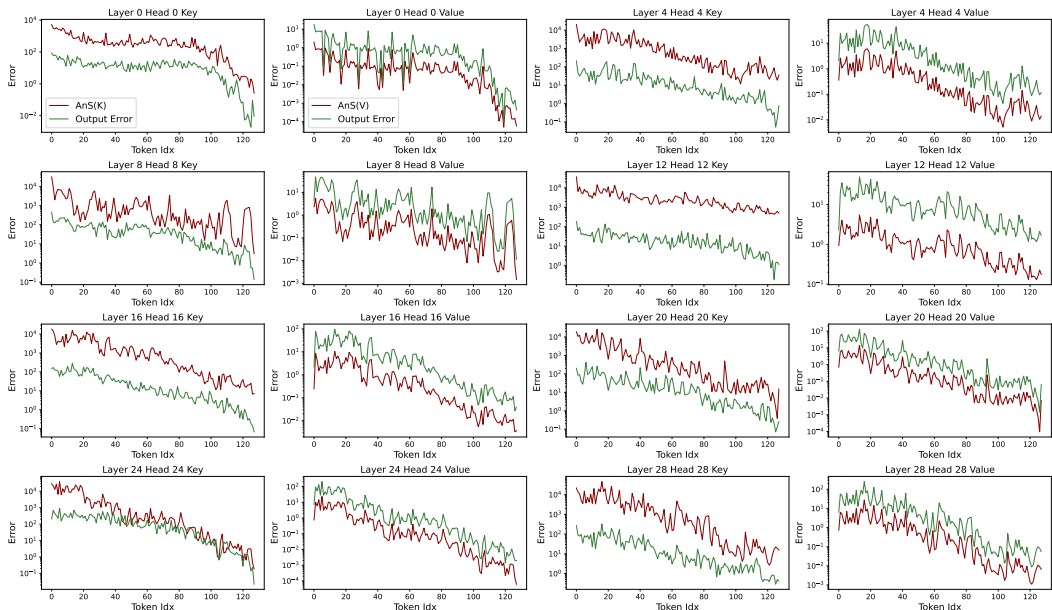

Figure 9: The effectiveness of AnS.

## E    ANS DISTRIBUTION DURING DECODING

To illustrate the distribution of AnS during decoding, we sampled prompts from Qasper within LongBench for visualization. As shown in Figure 10, we present the distribution of AnS($K$) and AnS($V$) across different layers and heads, specifically when decoding the first token. Our observations reveal that high AnS values during decoding are predominantly concentrated on adjacent tokens and at the attention sink tokens. Since sink tokens often lead to significant error propagation and can be dynamically identified by AnS during prefill, we simplify the design of AnS during decoding by employing a sliding window to ensure model performance.

## F    COMPUTATIONAL PROCEDURE OF ANS IN THE ONLINE STAGE

---
**Algorithm 1** The computation of AnS in the online stage.
___

1: **Input:** Query ($Q$), key ($K$), value ($V$)
2: **Output:** AnS of KV, i.e., AnS($K$) and AnS($V$)
3: $(O, L, M, \|Q_{i,:}\|_{L_2}) \leftarrow \texttt{FlashAttention}(Q, K, V)$
4: **for** each block key index $j$ in parallel (assigned to GPU block) **do**
5:    **for** each block query index $i$ **do**
6:       $S_{i,j} \leftarrow \langle Q_{h,i}, K_{h,j} \rangle$
7:       $A_{i,j} \leftarrow \exp(S_{i,j} - M_{h,i})/L_{h,i}$
8:       $\text{AnS}(K)_i \leftarrow \text{AnS}(K)_i + \texttt{col\_sum}(A_{i,j} \cdot (1 - A_{i,j}), \text{ row-wise})$
9:       $\text{AnS}(V)_i \leftarrow \text{AnS}(V)_i + \texttt{col\_sum}(A_{i,j}, \text{ row-wise})$
10:   **end for**
11: **end for**
12: **return** AnS($K$), AnS($V$)

---

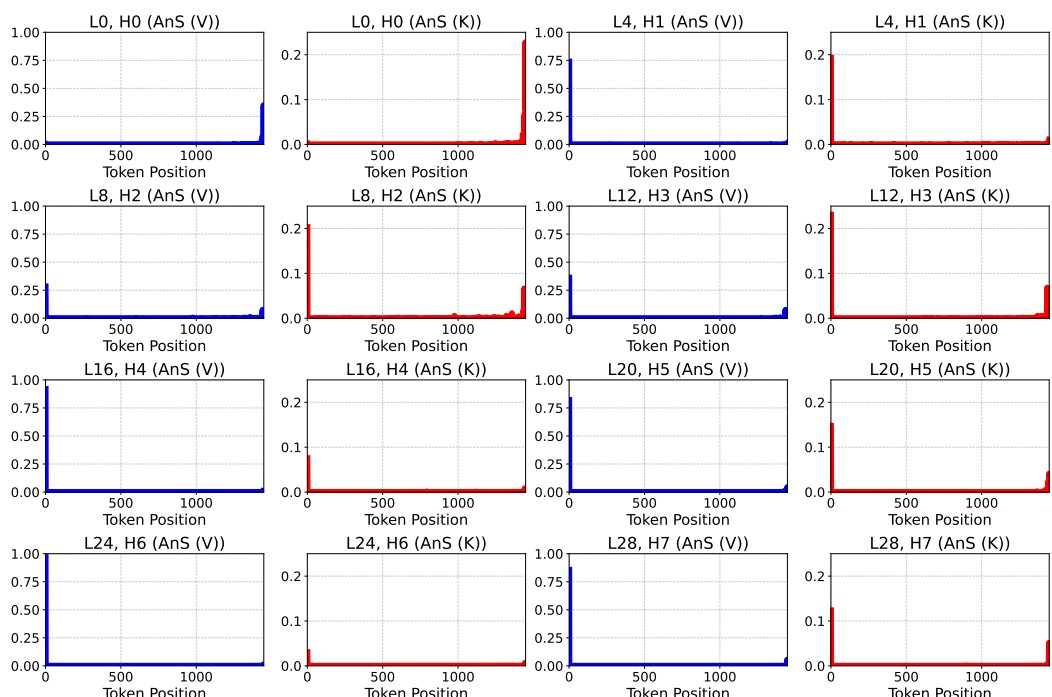

Figure 10: AnS Distribution on Sampled Prompts from Qasper Using LLaMA-3-8B-Instruct During First-Token Decoding.

## G CALIBRATION SET IMPACT

As shown in the Table 2, we observe that for VQ-based quantization in the ultra-low-bit regime, the calibration set significantly impacts the perplexity results. However, AnTKV with $1\%$ anchor tokens not only substantially reduces the PPL but also greatly mitigates the effect of different calibration sets.

Table 2: Perplexity experiment results on Mistral-7B, using the W2 and C4 training sets respectively as Calibration Sets. The "Vset" is the validation set related to W2 and C4. "Calib Set" represents "Calibration Set".

| Bits | Vset | 4 | | 2 | | 1 | | 0.75 | | 0.375 | |
|---|---|---|---|---|---|---|---|---|---|---|---|
| Calib Set | | W2 | C4 | W2 | C4 | W2 | C4 | W2 | C4 | W2 | C4 |
| Ours | W2 | 4.76 | 5.69 | 5.08 | 6.18 | 7.32 | 10.51 | 7.32 | 10.51 | 11.65 | 23.98 |
| | C4 | 4.79 | 5.69 | 5.32 | 6.15 | 10.14 | 10.09 | 10.90 | 10.80 | 24.16 | 19.95 |
| Ours-1% | W2 | 4.74 | 5.67 | 4.95 | 5.97 | 6.32 | 8.44 | 6.32 | 8.44 | 8.87 | 14.87 |
| | C4 | 4.75 | 5.66 | 5.02 | 5.94 | 7.13 | 8.13 | 7.79 | 8.56 | 12.87 | 13.07 |

To further investigate the impact of the calibration set on model performance, we used C4 as a calibration set to evaluate several subtasks within LongBench (qasper, trec, samsum, lcc, ropebench-p). As shown in the Table 3, we observed that there were some differences in the results of Trec and Repobench-p when using Wikitext-2 and C4, while the differences were not significant for the other tasks.

Table 3: Performance on LongBench Subtasks with Wikitext-2 (W2) and C4 Calibration Sets at Different Bits using LLaMA-3-8B-Instruct. "Calib Set" represents "Calibration Set", and "repobc-p" represents "repobench-p".

| Bits | 4 | | 2 | | 1 | | 0.75 | | 0.375 | |
|---|---|---|---|---|---|---|---|---|---|---|
| Calib Set | W2 | C4 | W2 | C4 | W2 | C4 | W2 | C4 | W2 | C4 |
| qasper | 40.46 | 39.98 | 39.04 | 38.2 | 25.95 | 26.51 | 25.48 | 25.49 | 22.41 | 23.27 |
| trec | 69.33 | 69.33 | 67 | 64.67 | 38.67 | 42.33 | 39.67 | 41.33 | 38 | 38 |
| samsum | 40.2 | 40.27 | 38.61 | 38.22 | 30.0 | 30.3 | 29.57 | 29.29 | 25.5 | 24.82 |
| lcc | 59.84 | 59.07 | 60.94 | 59.15 | 53.97 | 53,93 | 52.97 | 52.01 | 49.61 | 49.79 |
| repobc-p | 44.24 | 41.3 | 45.29 | 42.68 | 38.53 | 37.94 | 37.87 | 38.02 | 34.54 | 34.71 |

## H ANCHOR TOKENS NUMBER IMPACT

To investigate the impact of the number of anchor tokens on model performance, we conducted Perplexity evaluations on Mistral-7B and LLaMA-3-8B, both with a context length of 8192. We performed evaluations using no anchor tokens and with anchor token percentages of 1% (82), 2% (164), 5% (410), 10% (820), 15% (1230), and 20% (1640). The corresponding results are presented in Figures 11 and 12. For the 2-bit and 4-bit results, using $1\%$ of anchor tokens kept the error within 0.6 compared to FP16. However, for the 1-bit and sub-bit results, we needed to increase the number of anchor tokens to control the error within an acceptable range. Nevertheless, AnTKV provides a feasible technical pathway for ultra-low-bit quantization of the KV cache.

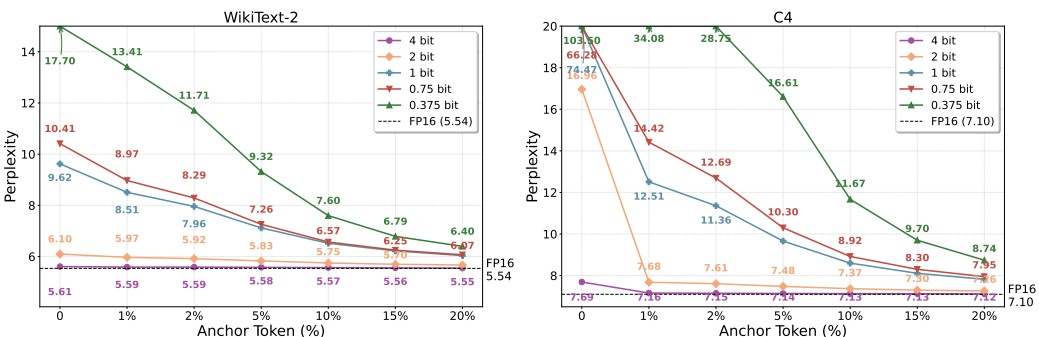

Figure 11: Perplexity results on LLaMA-3-8B with varying anchor token numbers.

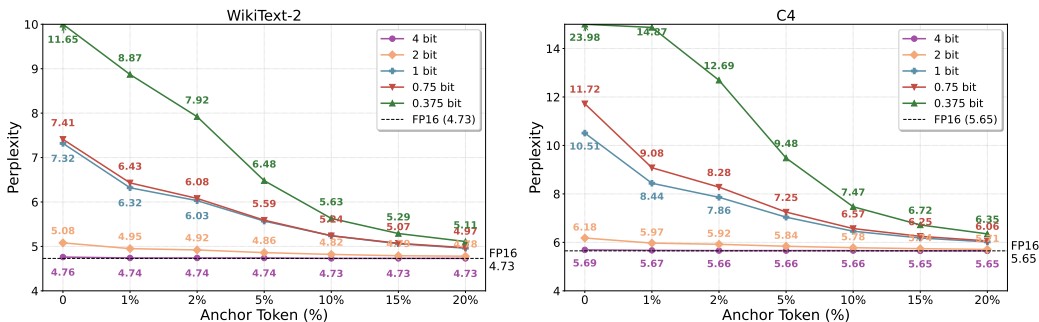

Figure 12: Perplexity results on Mistral-7B with varying anchor token numbers.

To further investigate the impact of anchor token numbers on downstream tasks, we evaluated different anchor token numbers on the Trec and Qasper subtasks of LongBench under ultra-low-bit

quantization settings. For convenience, we approximated $1\%$ of the anchor token number as 64. The results are shown in Figure 13. it illustrates that both the Trec and Qasper subtasks exhibit a consistent improvement pattern as the number of anchor tokens increases. In particular, moving from 0% to 1% anchor tokens leads to a substantial performance gain across all quantization settings, highlighting that even a very small proportion of anchor tokens can effectively mitigate the degradation introduced by ultra-low-bit quantization. Beyond this point, the improvements from 1% to 2%, 2% to 5%, and 5% to 10% follow an approximately linear trend, with performance gradually approaching the FP16 baseline. These results demonstrate that anchor tokens play a dual role that a small fraction is sufficient to deliver immediate and significant benefits, while larger allocations further provide steady, near-linear enhancements in downstream task performance.

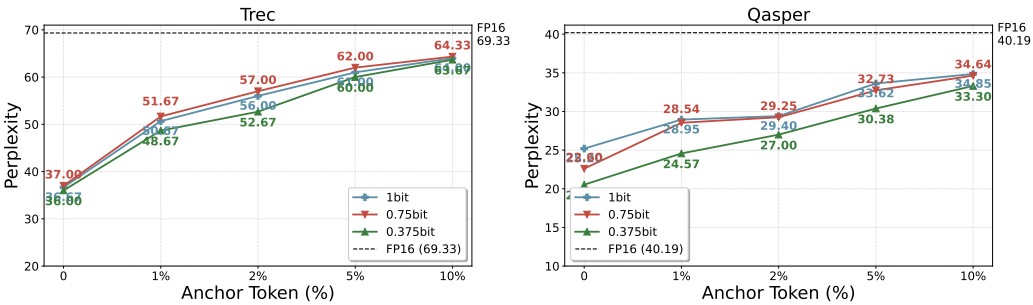

Figure 13: Trec and Qasper results on LLaMA-3-8B-Instruct with varying anchor token numbers.

# I  USE OF LLMS

In preparing this manuscript, we utilized ChatGPT-5 as a writing and editing assistant. Its role was limited to enhancing the clarity and fluency of the English in various sections. All scientific ideas, research methodology, experimental design, result analysis, and technical contributions are solely the product of the human authors.

