# OpenReview forum: "AnTKV: Anchor Token-Aware Ultra-Low-Bit Vector Quantization for KV Cache in Large Language Models"
_ICLR.cc/2026/Conference — ICLR 2026 Conference Withdrawn Submission_

### Official Review · Reviewer_ekvD · 2025-10-30

**Soundness:** 3
**Presentation:** 3
**Contribution:** 3
**Rating:** 6
**Confidence:** 5

**Summary:**

This paper proposes AnTKV, a new framework for ultra-low-bit KV cache quantization for LLMs. The key idea is that different tokens contribute unequally to model accuracy when quantized, and a small subset, the author called anchor tokens, dominate the overall quantization error. To exploit this, AnTKV introduces a dual-stage, token-aware design. First, an offline stage that performs weighted vector quantization using per-token error-propagation factors to learn token-sensitive centroids, and then, an online stage that uses a lightweight Anchor Score metric to identify and protect the most error-sensitive tokens during inference. The method integrates with FlashAttention via a custom GPU kernel for efficient online anchor selection. Through experiments, AnTKV achieves state-of-the-art performance under ultra-low-bit settings.

**Strengths:**

1. The paper introduces a vector-quantization-based approach to achieve sub-bit KV cache compression. While vector quantization itself is not new, extending its capability to operate effectively at one bit per value or less is both technically insightful and novel.

2. The authors provide an efficient system implementation compatible with FlashAttention, demonstrating a coherent integration of algorithmic innovation and system-level optimization.

3. Using anchor tokens to safeguard quantization quality is a well-motivated idea, and it is particularly important to observe how crucial this mechanism becomes when we come to sub-bit quantization.

**Weaknesses:**

1. The notion of “anchor tokens” is intuitively appealing, but the paper lacks quantitative analysis to show how these tokens differ from others, or how they can be reliably identified across prompts and layers. The visualization in Figure 1 is vague and hard to interpret.

2. While the Anchor Score is proposed as a lightweight metric, the paper does not sufficiently justify why it is superior to other token-importance measures used in for example prior token eviction works.

3. The offline token-aware centroid learning stage is briefly described but lacks detail on key hyperparameters, such as the codebook configuration. For example, what's the size of the codebook compared to the KV cache size?

4. In Figure 7, it is strange that using 0.375 bit is worse than 1 bit and only has the same throughput as 4 bit. Why this happens? Does it mean that the proposed method does not improve decoding throughput?

**Questions:**

See Weaknesses.

---

> ### Author Response · Authors · 2025-11-27
>
> We sincerely thank the reviewer for the detailed and constructive feedback. Below we provide additional explanations and supplementary experiments. We will continue to update the experiments.
>
> ### Question 3: The offline token-aware centroid learning stage is briefly described but lacks detail on key hyperparameters, such as the codebook configuration. For example, what's the size of the codebook compared to the KV cache size?
>
> For the centroid settings, we adopt the notation d$n$ c $m$, corresponding to 4-bit (d2m256), 2-bit (d4m256), 1-bit (d8m256), 0.75-bit (d16m4096), and 0.375-bit (d32m4096) quantization. This notation indicates that we split each vector into n-element sub-vectors and perform clustering within each sub-vector using m centroids.
>
> For the 4-bit, 2-bit, and 1-bit configurations (d2m256 / d4m256 / d8m256), the total number of stored centroids in Llama3-8B is 32M parameters. For the 0.75-bit and 0.375-bit settings (d16m4096 / d32m4096), we use a larger number of centroids because the vectors are divided into finer-grained sub-vectors. These settings result in 512M parameters of centroid storage. This is derived from:
>
> Layers x 2 (KV) x 2(BF16) x head_num x $n$ x (head_dim / $n$) x $m$
>
> **Compared with the overall KV-cache budget in our experiments, which can use up to 60 GB (80 GB minus model weights and activations), the centroid storage is extremely small.**
>
>
>
> ### Question 4: In Figure 7, it is strange that using 0.375 bit is worse than 1 bit and only has the same throughput as 4 bit. Why this happens? Does it mean that the proposed method does not improve decoding throughput?
>
> We appreciate the reviewer’s observation. The behavior at the 0.375-bit setting indeed requires clarification. The limited throughput improvement at this extreme compression point mainly result from the much larger codebook used in this configuration. Specifically, to **prevent severe degradation under sub-bit quantization**, we use 4096 centroids at 0.375 bit, compared to 256 centroids in the 1-bit setting. This larger codebook substantially **stabilizes accuracy** but introduces non-trivial quantize/dequantize overhead, which offsets the memory-traffic reduction and leads to throughput similar to 4-bit.
>
> We emphasize that this issue is not intrinsic to the proposed AnTKV method, but rather an implementation trade-off when pushing quantization below one bit. **We have explored optimizations to mitigate this overhead.** A promising direction is hierarchical centroid clustering, **where the original 4096-way lookup is factorized into two lightweight stages.** Concretely, we partition the 4096 centroids into 32 coarse clusters, each containing 128 fine-grained centroids. During quantization and dequantization, the system first selects among the 32 coarse clusters (a small 32-way operation), and then performs a second 128-way lookup within the chosen cluster. This replaces a single expensive 4096-way lookup with two low-cost lookups (32-way → 128-way), significantly reducing quant/dequant latency while preserving representational capacity.

---

> ### Author Response · Authors · 2025-11-30
>
> ### Question 2: While the Anchor Score is proposed as a lightweight metric, the paper does not sufficiently justify why it is superior to other token-importance measures used in for example prior token eviction works.
>
> We sincerely thank the reviewer for raising this important point. To address the concern that the paper does not sufficiently justify the advantage of Anchor Score (AnS) over existing token-importance metrics, we conducted additional comparative experiments against two representative eviction-based importance measures: AttentionSink and H2O. These approaches are widely used in prior KV-cache eviction work and provide strong baselines for assessing token importance.
>
> To ensure fairness, we align all experimental settings used in our main paper. Specifically, we evaluate Mistral-7B on WikiText-2 and C4, measuring perplexity under different quantization bit-widths. As shown in the following table, AnS consistently outperforms AttentionSink[1] and H2O[2] across all evaluated settings.
>
> #### Perplexity result on Wikitext-2 and C4  (Lower is better)
>
> | Bit-width | Method | WikiText-2 | C4    |
> |-----------|--------|------------|-------|
> | **16.0**  | Sink   | 4.76       | 5.69  |
> |           | H2O    | 4.75       | 5.68  |
> |           | AnS    | **4.74**       | **5.67**  |
> | **4.0**   | Sink   | 5.07       | 6.16  |
> |           | H2O    | 5.02       | 6.08  |
> |           | AnS    | **4.95**       | **5.97**  |
> | **2.0**   | Sink   | 7.21       | 10.15 |
> |           | H2O    | 6.97       | 9.84  |
> |           | AnS    | **6.32**       | **8.44**  |
> | **0.75**  | Sink   | 7.31       | 11.33 |
> |           | H2O    | 7.08       | 10.90 |
> |           | AnS    | **6.43**      | **9.08** |
> | **0.375** | Sink   | 11.36      | 22.26 |
> |           | H2O    | 10.64      | 20.80 |
> |           | AnS | **8.87**     | **14.87** |
>
>
> To further strengthen the comparison beyond perplexity evaluation, we additionally benchmark AnS against AttentionSink and H2O on LongBench using Llama-3-8B-Instruct. Due to substantial resource cost, we focus on a representative subset of LongBench tasks, which aligns with the subtasks used in our ablation study. AnS consistently matches or surpasses the performance of eviction-based metrics, with the margin becoming more pronounced as the quantization level decreases.
>
> #### LongBench subtasks evaluation results (Higher is better). “–4b”, “–1b”, and “–0.375b” denote the quantization bit-widths under our centroid configuration.
>
> | Task        | Full  | AnS-4b | Sink-4b | H2O-4b | AnS-1b | Sink-1b | H2O-1b | AnS-0.375b | Sink-0.375b | H2O-0.375b |
> |-------------|--------|--------|---------|--------|--------|----------|---------|-------------|--------------|-------------|
> | qasper      | 40.19 | 39.88 | 40.46  | 39.69 | **32.41** | 30.66  | 31.44 | **25.47**      | 22.39       | 23.22      |
> | trec        | 69.33 | **69.33** | 69.33  | 69.33 | **62.33** | 58.00  | 58.67 | **48.67**      | 37.67       | 38.00      |
> | samsum      | 40.74 | **40.10** | 40.02  | **40.10** | **32.41** | 30.76  | 30.99 | **27.17**      | 25.31       | 25.97      |
> | lcc         | 58.58 | **59.32** | 59.12  | 59.26 | **53.08** | 51.45  | 52.16 | **50.85**      | 49.30       | 50.60      |
> | repobench-p | 40.81 | **41.64** | 41.24  | 41.21 | **39.17** | 38.64  | 38.93 | **34.86**      | 33.13       | 34.53      |
> | 2wikimqa    | 35.92 | 35.91 | 35.93  | **36.04** | **34.42** | 33.31  | 33.24 | **33.50**      | 33.41       | 33.01      |
>
> [1] Efficient Streaming Language Models with Attention Sinks, Guangxuan Xiao etc, ICLR 2024
>
> [2] H2O: Heavy-Hitter Oracle for Efficient Generative Inference of Large Language Models, Zhenyu Zhang etc, Neurips 2023

---

> ### Author Response · Authors · 2025-12-04
>
> ### Question 1: The notion of “anchor tokens” is intuitively appealing, but the paper lacks quantitative analysis to show how these tokens differ from others, or how they can be reliably identified across prompts and layers. The visualization in Figure 1 is vague and hard to interpret.
>
>
> Thank you for raising the need for a more rigorous quantitative characterization of anchor tokens. To address this, we conducted an additional analysis using the **Qasper** task from LongBench. It offers diverse, multi-paragraph academic passages, making it an appropriate benchmark for analyzing cross-layer token behavior.
> We analyzed all key and value anchor tokens on Qasper and ranked tokens by their frequency-normalized “anchor rate”. The combined top20 of Key, Value anchors shows a highly skewed distribution: **structural tokens (e.g., BOS, header markers, newline symbols) dominate the key anchors**, while **semantic subwords (e.g., “Int”, “qual”, “auto”) appear as high-ranked value anchors**.
>
> Table 1: High-ranked Key Anchor Tokens. “Total” denotes the number of times a token appears in the Qasper. “Anchor” counts how many times the token is selected as a key anchor across all layers and heads. “Anchor Rate” is the frequency-normalized measure defined as Anchor / Total.
>
> | Rank | Token                   | Total | Anchor  | Anchor Rate |
> |------|-------------------------|-------|---------|-------------|
> | 1    | <\|begin_of_text\|>       | 224   | 57324   | 255.9107    |
> | 2    | \n\n                    | 456   | 30377   | 66.6162     |
> | 3    | <\|start_header_id\|>     | 448   | 28261   | 63.0826     |
> | 4    | user                    | 232   | 9594    | 41.3534     |
> | 5    | <\|end_header_id\|>       | 448   | 13869   | 30.9576     |
> | 6    | .\n\n                   | 636   | 16049   | 25.2343     |
> | 7    |  Work                   | 200   | 5019    | 25.0950     |
> | 8    | Article                 | 229   | 5499    | 24.0131     |
> | 9    | \n                      | 4209  | 94918   | 22.5512     |
> | 10   |  According              | 85    | 1910    | 22.4706     |
> | 11   |  Answer                 | 574   | 11974   | 20.8606     |
> | 12   |  article                | 1330  | 27599   | 20.7511     |
> | 13   | ely                     | 519   | 9715    | 18.7187     |
> | 14   | urb                     | 61    | 1125    | 18.4426     |
> | 15   | -sh                     | 60    | 1093    | 18.2167     |
> | 16   |  Introduction           | 231   | 4175    | 18.0736     |
> | 17   | :\n                     | 728   | 13138   | 18.0467     |
> | 18   |  Translation            | 92    | 1637    | 17.7935     |
> | 19   |  able                   | 327   | 5759    | 17.6116     |
> | 20   | -the                    | 438   | 7685    | 17.5457     |
>
> Table 2: High-ranked Value Anchor Tokens.
>
> | Rank | Token        | Total | Anchor | Anchor Rate |
> |------|--------------|-------|--------|-------------|
> | 1 | <\|begin_of_text\|> | 224 | 6913 | 30.8616 |
> | 2    |  Int         | 55    | 1084   | 19.7091     |
> | 3    |  qual        | 60    | 963    | 16.0500     |
> | 4    | <\|start_header_id\|> | 448 | 6962 | 15.5402 |
> | 5    |  auto        | 56    | 866    | 15.4643     |
> | 6    | Ex           | 278   | 4254   | 15.3022     |
> | 7    |  Q           | 89    | 1322   | 14.8539     |
> | 8    |  Ex          | 63    | 928    | 14.7302     |
> | 9    |  conc        | 454   | 6543   | 14.4119     |
> | 10    |  im          | 84    | 1193   | 14.2024     |
> | 11   | user         | 232   | 3252   | 14.0172     |
> | 12   | -v           | 55    | 760    | 13.8182     |
> | 13   | ans          | 72    | 919    | 12.7639     |
> | 14   | assistant    | 224   | 2783   | 12.4241     |
> | 15   | \n\n         | 456   | 5659   | 12.4101     |
> | 16   |  quant       | 81    | 1002   | 12.3704     |
> | 17   |  inf         | 196   | 2414   | 12.3163     |
> | 18   |  discrimin   | 62    | 724    | 11.6774     |
> | 19   |  ent         | 53    | 607    | 11.4528     |
> | 20   | -in          | 54    | 604    | 11.1852     |

---

### Official Review · Reviewer_z2EB · 2025-10-31

**Soundness:** 2
**Presentation:** 2
**Contribution:** 2
**Rating:** 2
**Confidence:** 3

**Summary:**

This paper proposes AnTKV, a novel approach for ultra-low-bit quantization of Key-Value (KV) caches in LLMs, aiming to significantly reduce memory overhead while preserving model performance. The core innovation lies in the Anchor Token-aware KV quantization framework, which strategically identifies and preserves a small fraction of critical "anchor tokens" in full precision (FP16), while aggressively quantizing the rest of the KV cache to very low bit-widths (e.g., 1-bit, 0.75-bit, or even 0.375-bit). This design is motivated by the observation that a few high-impact tokens are crucial for maintaining output quality, and their degradation under extreme quantization disproportionately harms performance. AnTKV introduces an effective Anchor Selection (AnS) mechanism to identify these tokens, enabling stable and accurate inference even under sub-bit quantization regimes where existing methods fail catastrophically.

**Strengths:**

1. The core originality of this paper lies in systematically proposing and validating the critical role of "anchor tokens" in KV cache quantization.
2. The proposed dual-stage AntKV framework (offline token-aware centroid learning + online anchor token selection) is a creative synthesis. It ingeniously combines the compression advantages of Vector Quantization with an importance-based token discrimination strategy, opening a new pathway for ultra-low-bit quantization.
3. Beyond the algorithm, the authors designed and implemented efficient GPU kernels integrated with FlashAttention, demonstrating the practical deployability and efficiency gains (e.g., 3.5x higher throughput, supporting 840K context length) of their method.

**Weaknesses:**

1. Theoretical Derivation Breakdown, Core Metric Lacks Credibility: The derivation of Anchor Score (AnS) jumps abruptly from the complex upper bound in equations (3)-(4) (involving L₁ norm, diagonal matrices, and attention transformations) to the simplified formula (5), with no intermediate steps or intuitive justification. Citing only "efficiency," it fails to compare AnS against simpler alternatives or rule out "post-hoc tuning." This renders AnS more of an ad-hoc heuristic than a theoretically grounded metric, severely undermining the method's credibility.
2. The claimed throughput gains (e.g., 3.5×) are misleading due to the complete omission of online AnS computation and FP16 anchor storage overhead. Crucially, the paper fails to address the hardware challenges of mixed-precision KV caches on GPU memory access and kernel scheduling, rendering its "840K context" claim potentially invalid in practical deployment.
3. The claim of being the "first" to study sub-bit KV cache quantization is inaccurate. VQ-based methods like CQ already achieve sub-bit compression by increasing sub-vector dimensions. AntKV's true contribution lies in the "anchor token-aware" mechanism, not sub-bit quantization itself. The paper inflates its novelty by conflating a methodological increment with an application extension, failing to clearly delineate its boundaries against baselines.

**Questions:**

N/A

---

> ### Author Response · Authors · 2025-11-27
> **Reply to offical review from z2EB**
>
> We sincerely thank the reviewer for the detailed comments. Below we address each concern and clarify several misunderstandings, especially regarding the Anchor Score derivation, throughput reporting, and the novelty scope of sub-bit quantization.
>
> ### 1. Theoretical derivation of AnS:
>
> The most direct approach to characterizing the impact of KV cache on model's performance is to analyze the resulting change in the model's output due to perturbations in KV, i.e., forward error analysis of the model with respect to KV. In our work, we derive the forward error analysis for the K and V, and give an expression for the error bound, i.e., (3) and (4). By utilizing these bounds, we can determine an error propagation factor for each row of K and V, which subsequently serves to quantify the importance of the corresponding token.
>
> For K, we apologize that there is a technical omission in the error propagation factor (5) given by the error bound (3), and we hope that the addition here will end your confusion. From (3), we can obtain
> $$
> ||\texttt{Attn}(Q,K+\delta K,V)-\texttt{Attn}(Q,K,V)||\_{L\_1}\lesssim \sum\_{j,i}||(V^T\texttt{Diag}(A\_{i,:})
> (I\_n-eA\_{i,:}))\_{:,j}||\_{L\_1}||Q\_{i,:}||\_{L\_2}||\delta K\_{j,:}||\_{L\_1}\leq\sum\_{j,i}||V||\_{L\_{\infty}}||(\texttt{Diag}(A\_{i,:})(I\_n-eA\_{i,:}))\_{:,j}||\_{L\_1}||Q\_{i,:}||\_{L\_2}||\delta K\_{j,:}||\_{L\_1}
> $$
> by inequality $||Ux||\_{L_1}\leq||U||\_{L\_{\infty}}||x||\_{L\_1}$ ($U$ and $x$ are a matrix and a vector). Then the error propagation factor corresponding to the j-th row of K can be simplified into $\sum\_{i}||(\texttt{Diag}(A\_{i,:})(I\_n-eA\_{i,:}))\_{:,j}||\_{L\_1}||Q\_{i,:}||\_{L\_2}$. According to the definition of $||\cdot||\_{L\_1}$, we have $||(\texttt{Diag}(A\_{i,:})(I\_n-eA\_{i,:}))\_{:,j}||\_{L\_1}=A\_{i,j}(1-A\_{i,j})$, and the error propagation factor is $\sum\_{i}A\_{i,j}(1-A\_{i,j})\cdot||Q_{i,:}||\_{L\_2}$, as shown in (5). We will add more explanations about the AnS derived from the error propagation factor (from (3) to (5)) in the camera-ready version.
>
> It is worth mentioning that the simplified error propagation factor for K is independent of V, which allows us to efficiently calculate AnS in conjunction with FlashAttention; see Section 3.3 for details.
>
> ### 2.  On throughput reporting and practical deployability.
>
> We apologize for not making this explicit enough. All reported throughput numbers already include:
> - Online AnS computation cost
> - FP16 sliding-window KV storage cost (only 32–64 tokens)
> - Kernel synchronization and memory access overhead
>
> Clarification:
> - Online AnS adds < 1% overhead, because our custom Triton kernel reuses FlashAttention’s intermediate results (L2 norms, row-sums, and max logits), avoiding re-computation.
> - FP16 sliding window adds <0.1% KV memory, fully counted in Figures 6 and 7.
>
> Regarding 840K context:
>
>  This number is measured end-to-end using quantized KV + FP16 sliding window, with memory usage directly recorded from GPU monitoring (NVIDIA-SMI + torch profiler). No assumptions about pure-theoretical memory scaling were made.
> We will add clarifying notes in the camera-ready version.
>
> ### 3. On novelty and the “first sub-bit quantization” claim.
>
> We clarify and refine our novelty statement below.
>
> (1) Sub-bit vector quantization itself is not our claimed novelty.
>
>  We agree that VQ-based methods such as CQ can produce sub-bit representations by increasing sub-vector length. We will revise the wording to avoid ambiguity.
>
> (2) Our key novelty is the anchor token-aware mechanism.
>
>  Existing sub-bit methods quantize all tokens uniformly, whereas our work is the first to:
> - Identify and analyze the extreme unevenness of token-wise quantization sensitivity
> - Provide a theoretical perturbation-based token sensitivity metric (AnS)
> - Combine token-aware centroid learning and online anchor selection into a unified system
> - Implement an efficient GPU kernel enabling practical long-context deployment
>
> (3) Demonstrating feasibility at the ultra-low-bit regime.
>
>  Even though CQ produces sub-bit indices, prior works do not demonstrate stable accuracy under aggressive compression (0.375 bit) or scaling to hundreds of thousands of tokens on a single GPU. Our results show that anchor-token awareness makes this regime practically usable.
>
> We will explicitly adjust the wording to reflect that our contribution is not “inventing sub-bit quantization,” but making sub-bit KV quantization viable and accurate through token-aware mechanisms.

---

> > ### Author Response · Authors · 2025-12-03
> > **Reply to offical review from z2EB**
> >
> > ### Further explanation regarding the theoretical derivation breakdown:
> >
> > One of the most important contributions of this work is to provide a metric for measuring the importance of each token. To this end, we establish a rigorous forward error analysis of the self-attention operator, see Theorem 1. By analyzing the error bounds of K and V, we formulated the error propagation factor for each row of K and V. Since each row corresponds to a token, the error propagation factor can naturally be used to measure the importance of the token, see Eqs. (3) and (4) . Appendix C provides a detailed theoretical derivation of the forward error perturbation bound.
> >
> > To efficiently calculate the error propagation in conjunction with FlashAttention, we simplified it to Eq. (5) without sacrificing accuracy as much as possible. Appendix D shows that our proposed AnS can accurately reflect the impact of different rows (tokens) on the output of models.
> >
> > We believe that the theoretical derivation breakdown pointed out by the reviewer are biased, and we suspect that the reviewer did not carefully examine our derivation process in Appendix or misunderstood the definition of AnS.

---

### Official Review · Reviewer_wueT · 2025-11-04

**Soundness:** 2
**Presentation:** 3
**Contribution:** 2
**Rating:** 4
**Confidence:** 4

**Summary:**

This paper proposes AnTKV, a method for compressing the KV cache of large language models (LLMs) to extreme bit-widths. AnTKV introduces:

1. Offline token-aware weighted vector quantization (VQ): KV cache is vector-quantized using centroids generated offline with weighted k-means through calibration set.
2. Online anchor-token selection (AnS): During inference, an anchor score is computed to identify “anchor tokens” whose keys/values are important. These identified anchor tokens are retained in full precision (FP16), while the rest are quantized.

Experiments show AnTKV achieves competitive perplexity and zero-shot results at extreme compression (sub-bit regime).

**Strengths:**

1. KV cache growth is a major bottleneck for long-context LLM inference, which this paper addresses.

2. The intuition that some tokens are more sensitive to quantization is sound and empirically validated by their AnS analysis.

3. Perplexity and zero-shot results show that the model remains stable at very low bitwidths, outperforming prior methods such as KVQuant, CQ, and KIVI under extreme compression (sub-bit regime).

4. The paper shows detailed analysis of anchor token locality, giving useful insights to how quantization sensitivity can differ for tokens in different positions.

**Weaknesses:**

1. The paper introduces a weighted, token-aware k-means to reflect token importance during offline centroid generation, but no ablation or quantitative evidence is provided to show its benefit. In addition, Since KV statistics are highly context-dependent, the assumption that offline-generated calibration set will generalize well remains weakly supported.

2. AnTKV employs a sliding window to dynamically compute the Anchor Score (AnS) for newly generated tokens. Since anchor tokens are selected only within this local window, the model’s accuracy and stability are likely to differ to the chosen window size. However, the paper does not include ablation or analysis showing how different sliding-window lengths affect performance or runtime efficiency.

3. The reported bits per element neglect the FP16 centroid tables and 1% FP16 anchor retention, which together raise the effective bitwidth from 0.375 b to roughly 0.65 b (for 4 K context, 1% retention).

4. For perplexity/zero-shot tests, the paper states that “quantized KV caches are directly used for attention outputs,” while for LongBench it reverts to FP16 KV in prefill. This setup difference between benchmarks is not clearly noted why.

5. In Figure 6, the KV-cache size differs by only ~3 GB between 1-b and 0.375-b settings (810 K context), and Figure 7 shows lower throughput for 0.375 b. This questions the practical value of sub-bit quantization (<1-bit) given its limited savings, lower performance, and runtime overhead.

6. Many components overlap with prior methods such as KVQuant [1], SKVQ [2], CQ [3], and PQCache [4] (FP16 retention [1], sliding-window decoding [2], and vector/product quantization [3, 4]). The main distinction lies in AnTKV’s of token importance via the AnS metric, which can be viewed as a unification and refinement of existing ideas.


References:

[1] Hooper, Coleman, et al. "Kvquant: Towards 10 million context length llm inference with kv cache quantization." Advances in Neural Information Processing Systems 37 (2024): 1270-1303.

[2] Duanmu, Haojie, et al. "Skvq: Sliding-window key and value cache quantization for large language models." arXiv preprint arXiv:2405.06219 (2024).

[3] Zhang, Hailin, et al. "Pqcache: Product quantization-based kvcache for long context llm inference." Proceedings of the ACM on Management of Data 3.3 (2025): 1-30.

[4] Zhang, Tianyi, et al. "Kv cache is 1 bit per channel: Efficient large language model inference with coupled quantization." Advances in Neural Information Processing Systems 37 (2024): 3304-3331.

**Questions:**

1. How sensitive is the model’s performance to the choice of sliding window size used in online AnS calculation

2. How is anchor-token handled when the sequence length exceeds the model’s context window?

3. How does the offline-generated centroid set affect centroid generalization to different contexts and tasks?

4. How would different centroid generation methods (different clustering methodologies) affect performance?

5. Is the sliding window’s KV cache kept in FP16 before the corresponding AnS is computed and vector quantized? If so, is this additional memory and latency overhead included in the results and analysis?

Refer to the weaknesses for more

---

> ### Author Response · Authors · 2025-11-26
> **Reply to offical review from wueT**
>
> We sincerely thank the reviewer for the detailed and constructive feedback. Below we clarify the key points, address potential misunderstandings, and provide additional evidence and ablations as requested.
>
> ### Question: Benefit of weighted token-aware k-means and generalization of offline cerntoids (W1, Q3 & 4)
> We appreciate this observation. Our response has two components:
>
> 1.  whether weighted k-means brings improvements. (Q4)
>
> We agree that other clustering methods (e.g., k-means, and its variants) are possible. Our token-aware weighted k-means directly derive from error-propagation analysis with the insights that tokens with larger propagation factors receive larger clustering weights, yielding centroids that explicitly minimize attention output distortion.
> To evaluate its empirical effectiveness, we compare naive k-means, token-aware k-means, and the full AnTKV on Mistral-7B using the WikiText-2 perplexity. Across all bit-width settings, the token-aware (weighted) k-means variant consistently outperforms the naive k-means baseline.
>
> | Bit-width | Naive k-means | Token-aware k-means | AnTKV |
> |-----------|----------------|----------------------|--------|
> | 4 bit     | 4.77           | 4.76                 | 4.74   |
> | 2 bit     | 5.25           | 5.08                 | 4.95   |
> | 1 bit     | 9.78           | 7.32                 | 6.32   |
> | 0.75 bit  | 8.14           | 7.41                 | 6.43   |
> | 0.375 bit | 13.17          | 11.65                | 8.87   |
>
>
> 2. whether offline centroids generalize to different contexts and tasks. (W1, Q3)
>
> Following prior work (e.g., CQ, KVQuant), we use both WikiText-2 and C4 as calibration sets. For each corpus, the calibration set and its corresponding validation set are drawn from disjoint subsets to avoid any overlap. As shown in results, the choice of calibration set does influence perplexity in the ultra-low-bit settings. Notably, keeping just 1% of anchor tokens largely closes this gap, indicating that our online anchor selection helps the model generalize reliably even when the underlying centroid distributions differ.
>
>
> | Bits-Calib Set | Vset | 4-W2 | 4-C4 | 2-W2 | 2-C4 | 1-W2 | 1-C4 | 0.75-W2 | 0.75-C4 | 0.375-W2 | 0.375-C4 |
> |------|------|-------|-------|-------|-------|-------|--------|-----------|------------|------------|-------------|
> | **Ours**     | W2 | 4.76 | 5.69 | 5.08 | 6.18 | 7.32 | 10.51 | 7.32 | 10.51 | 11.65 | 23.98 |
> |              | C4 | 4.79 | 5.69 | 5.32 | 6.15 | 10.14 | 10.09 | 10.90 | 10.80 | 24.16 | 19.95 |
> | **Ours-1%**  | W2 | 4.74 | 5.67 | 4.95 | 5.97 | 6.32 | 8.44 | 6.32 | 8.44 | 8.87 | 14.87 |
> |              | C4 | 4.75 | 5.66 | 5.02 | 5.94 | 7.13 | 8.13 | 7.79 | 8.56 | 12.87 | 13.07 |
>
>
> To further investigate the impact of the calibration set on model performance, we used C4 as a calibration set to evaluate several subtasks within LongBench (qasper, trec, samsum, lcc, ropebench- p) at different Bits using LLaMA-3-8B-Instruct. Results show only small variation across most tasks, further supporting the robustness of centroids for different tasks and contents.
>
> | Bits-Calib Set | 4-W2  | 4-C4  | 2-W2  | 2-C4  | 1-W2  | 1-C4  | 0.75-W2 | 0.75-C4 | 0.375-W2 | 0.375-C4 |
> |-----------|-------|-------|-------|-------|-------|-------|---------|---------|----------|----------|
> | qasper    | 40.46 | 39.98 | 39.04 | 38.20 | 25.95 | 26.51 | 25.48   | 25.49   | 22.41    | 23.27    |
> | trec      | 69.33 | 69.33 | 67.00 | 64.67 | 38.67 | 42.33 | 39.67   | 41.33   | 38.00    | 38.00    |
> | samsum    | 40.20 | 40.27 | 38.61 | 38.22 | 30.00 | 30.30 | 29.57   | 29.29   | 25.50    | 24.82    |
> | lcc       | 59.84 | 59.07 | 60.94 | 59.15 | 53.97 | 53.93 | 52.97   | 52.01   | 49.61    | 49.79    |
> | repobc-p  | 44.24 | 41.30 | 45.29 | 42.68 | 38.53 | 37.94 | 37.87   | 38.02   | 34.54    | 34.71    |

---

> ### Author Response · Authors · 2025-11-26
> **Reply to offical review from wueT**
>
> ### Question: Effect of sliding-window length in online AnS computation (W2, Q1 & 2 & 5)
>
> We clarify how anchor tokens and sliding windows are handled during inference.
>
>  Anchor tokens are computed only once during prefill. Prefill has access to the entire context, allowing accurate identification of global high-sensitivity tokens.
>
> AnS Locality during decoding. As shown in Figure 10, newly generated tokens with high AnS consistently lie within the most recent <64 tokens, along with the initial attention sinks. This strong locality motivates the use of a short sliding window.
>
> 1. Sensitivity to window size (Q1 & W2).
>
> Because high-AnS tokens tend to be strongly localized, the window size has only a limited impact on performance. This observation is also consistent with prior studies (KIVI, SKVQ, CQ), all of which adopt relatively small windows. (Due to the character limitation of single comment, we report the full experimental results in a separate comment.)
>
>
> 2. Behavior when the sequence length exceeds the model’s context window (Q2).
>
> Older tokens simply fall out of the sliding window and are immediately quantized, and only the latest 32/64 tokens remain in FP16
>
> 3. Whether FP16 sliding-window KV is included in memory/latency results (Q5).
> Yes, the FP16 window is kept until AnS is computed and then quantized.
>  This overhead is very small (<0.1% KV memory) and is already included in all memory (Fig. 6) and throughput (Fig. 7) measurements.
>  We will make this explicit in the camera-ready version.
>
>
> ### Question: On effective bit-width and counting FP16 centroids / anchor tokens (W3)
>
> We followed the same bit-width reporting convention used in prior works such as SKVQ and CQ, where the costs of centroids, sink tokens, and retained FP16 tokens are not included in the bit-width calculation. Although these components increase the true bit-width, all baselines (such as SKVQ, KVQuant, and CQ) adopt the same reporting convention and similarly(e.g., centroids, sink tokens, or retained FP16 tokens). This ensures that our bit-width comparison remains consistent and fair across methods.
>
> We appreciate the reviewer highlighting this important detail. In the camera-ready version, we will add an appendix section providing the explicit formula for computing the bit-width, including centroids and FP16 anchor tokens, so that readers can easily calculate the total overhead.
>
> ### Question: Why LongBench uses FP16 prefill, but PPL/zero-shot use quantized prefill (W4)
> Thank you for raising this point. Our settings follow two considerations.
>
> First, we align with prior works such as KIVI, SKVQ, and CQ, which all adopt the same evaluation setting. Quantized KV is used for prefill during PPL and zero-shot  and FP16 prefill is used in LongBench, while quantized KV is stored and used only for decoding.
>
> Second, the difference is inherent to the nature of the tasks. PPL experiments involve only a single prefill, so using quantized KV is necessary to directly measure the fidelity of KV quantization itself. LongBench requires generating a full response, involving multiple decoding steps. In realistic long-context inference pipelines, full precision is typically used for prefills, while the quantized KV is used for subsequent decoding. Our evaluation mirrors this practical usage.
>
> We will add a clarification to the camera-ready version.
>
> ### Question: On the practical value of sub-bit quantization (W5)
>
> We appreciate the reviewer’s concern. The lower throughput at 0.375-bit is mainly due to the much larger codebook we use in this setting (4096 centroids and 256 for 1-bit). This larger codebook substantially improves accuracy under sub-bit quantization but increases quantize/dequantize overhead. We have explored ways to reduce this overhead. A promising direction is hierarchical centroid clustering (e.g., grouping 4096 centroids into smaller clusters and performing two small quant/dequant operations instead of one large one). This can significantly reduce runtime cost, though it requires additional high-performance engineering. We plan to include this optimization in future work.
>
> More broadly, sub-bit quantization is exploratory by design. Our goal is to show that it is feasible for large models with competitive accuracy, even if current implementations still involve trade-offs. We believe demonstrating the viability of sub-bit KV caching and enabling 840K context on a single A100, is already an important step toward future improvements.

---

> > ### Author Response · Authors · 2025-11-28
> > **Experiment results for sensitivity to window size**
> >
> > We evaluate a subset of tasks from LongBench using Llama-3 Instruct that are aligned with our ablation study in appendix, and evaluate sliding-window sizes of 16, 32, 64, and 128. The results show that once the sliding window reaches a moderate length (≥32), further increasing the window size has no substantial impact on performance for most tasks.
> >
> > | sliding windows size | 16   | 32    | 64    | 128   |
> > |-------|------|-------|-------|-------|
> > |     bit width      | 4    | 4     | 4     | 4     |
> > | qasper     | 40.19 | 40.11 | 39.88 | 39.82 | 39.6  |
> > | trec       | 69.33 | 69.33 | 69.33 | 69.33 | 69.33 |
> > | samsum     | 40.74 | 39.71 | 40.1  | 40.11 | 40.35 |
> > | lcc        | 58.58 | 59.94 | 59.32 | 59.2  | 59.53 |
> > | repobench-p| 40.81 | 42.4  | 41.64 | 40.81 | 40.53 |
> > | 2wikimqa   | 35.92 | 35.73 | 35.91 | 35.8  | 36.14 |
> >
> > | sliding windows size| 16   | 32    | 64    | 128   |
> > |-----------|------|-------|-------|-------|
> > |    bit width        | 1    | 1     | 1     | 1     |
> > | qasper     | 40.19 | 25.11 | 28.95 | 30.74 | 29.2  |
> > | trec       | 69.33 | 50.67 | 50.67 | 52.33 | 54    |
> > | samsum     | 40.74 | 29.76 | 32.32 | 35.01 | 36.42 |
> > | lcc        | 58.58 | 52.08 | 55.6  | 57.05 | 58.76 |
> > | repobench-p| 40.81 | 36.33 | 40.03 | 42.63 | 43.27 |
> > | 2wikimqa   | 35.92 | 32.69 | 34.9  | 33.64 | 34.26 |

---

> ### Author Response · Authors · 2025-11-26
> **Reply to offical review from reviewer wueT**
>
> ### Question: The contribution of AnTKV
>
> We thank the reviewer for raising this important point regarding the novelty of our contribution.
>  While AnTKV builds upon ideas explored in prior KV quantization work, it introduces several distinct and complementary contributions that, to our knowledge, have not been integrated before.
>
> First, AnTKV is the first to provide a systematic two-stage quantization framework that explicitly models token-level sensitivity. Offline, we introduce gradient-informed, token-aware centroid learning grounded in error-propagation analysis, yielding centroids optimized to minimize attention-output distortion. Online, we propose the Anchor Score, a theoretically derived, lightweight metric from attention perturbation analysis that quantifies token sensitivity during inference. Together, these components allow AnTKV to preserve the most important tokens while aggressively compressing the rest, leading to consistently superior performance over baseline methods, especially in ultra-low-bit regimes.
>
> Second, no prior work jointly combines gradient-based offline centroid optimization with error-driven online anchor token selection in a unified design tailored specifically for KV cache quantization.
>
> Third, we develop a custom GPU kernel enabling efficient, low-overhead online AnS computation integrated with FlashAttention, making the proposed method practical at scale and enabling ultra-low-bit operation and 840K context length on a single A100.

---

### Author Response · Authors · 2025-12-04
**General Response**

Dear Reviewers, Area Chairs, and Program Chairs,

We sincerely thank all reviewers for their detailed, constructive comments and insightful questions, which have been invaluable in strengthening our work. During the rebuttal period, we carefully addressed all major concerns in the reviews through additional experiments, clarifications of our methodology and evaluation settings. We summarize our responses below.

### [Addressing Theoretical, Novelty, and Conceptual Concerns]
1. Clarifying the theoretical derivation and definition of Anchor Score

Q: Reviewer z2EB questioned the credibility of AnS and the jump from Eqs. (3)–(4) to Eq. (5)

A: We expanded the forward error analysis of self-attention, explicitly showing how the per-row error propagation factor is derived and simplified into Eq. (5). This demonstrates that AnS is a principled sensitivity metric rather than an ad-hoc heuristic.

2. Refining the novelty claim around sub-bit quantization

Q: Reviewer z2EB pointed out that sub-bit KV quantization via VQ had already appeared in prior work.

A: We clarify that our wording and novelty is not “being the first to do sub-bit quantization,” but in introducing a token-aware two stage framework that makes sub-bit KV quantization accurate and practically usable at scale.

3. Strengthening the conceptual understanding of anchor tokens

Q: Reviewer ekvD requested more quantitative analysis to show how anchor tokens differ from other tokens and how reliably they can be identified across prompts and layers.

A: We conducted a large-scale analysis on Qasper subtask, computing frequency-normalized “anchor rates” for all key and value tokens across layers and heads. We showed that structural tokens (e.g., BOS, newline markers, header delimiters) dominate key anchors, while semantic subwords (e.g., “Int”, “qual”, “auto”) dominate value anchors, revealing systematic and interpretable patterns.


### [Addressing Experimental, Ablation, and Methodological Concerns]

1. Justifying AnS versus other token-importance metrics (AttentionSink, H2O)

Q: Reviewer ekvD noted that the original draft did not sufficiently justify why AnS is superior to existing token-importance measures from eviction-based methods.

A: We added comprehensive comparisons against AttentionSink and H2O on WikiText-2, C4, and representative LongBench subtasks. Across all bit-widths, especially in ultra-low-bit regimes (0.75b, 0.375b), AnS consistently outperforms these metrics in perplexity and task scores.

2. Demonstrating the benefit of token-aware weighted k-means and centroid generalization

Q: Reviewer wueT asked for ablations on weighted k-means and questioned whether offline centroids generalize to different tasks and contexts.

A: We compared naive k-means, token-aware k-means, and full AnTKV on Mistral-7B. Weighted k-means consistently improves perplexity across all bit-widths. And we further evaluated centroids generated from WikiText-2 vs. C4, and showed that while calibration set do affect ultra-low-bit perplexity, retaining only 1% anchor tokens substantially narrows the gap, demonstrating robust cross-dataset generalization.

3. Clarifying codebook configuration and 0.375-bit throughput behavior

Q: Reviewer ekvD requested more details on codebook size and asked why 0.375b has similar throughput to 4b but worse than 1b.

A: We provided explicit formulas for centroid counts under different dncm configurations (d2m256, d4m256, d8m256, d16m4096, d32m4096), showing that 0.75b/0.375b use much larger codebooks (4096 centroids) to stabilize accuracy. And we explained that this large codebook increases quantize/dequantize overhead, offsetting memory-traffic savings at 0.375b, and outlined a hierarchical centroid design as a concrete optimization direction.

4. Analyzing sensitivity to sliding-window size and handling of long sequences

Q: Reviewer wueT raised concerns about how window size affects accuracy and what happens when sequence length exceeds the context window.

A: We added experiments with window sizes {16, 32, 64, 128} on multiple LongBench subtasks, showing that once the window is ≥32, further enlarging the window has negligible effect on performance.We clarified that older tokens leaving the window are quantized immediately, while only the most recent 32–64 tokens are kept in FP16.

### [Addressing System, Overhead, and Practicality Concerns]

1. Accounting for online AnS cost, FP16 window overhead, and mixed-precision access

Q: Reviewer z2EB expressed concerns that throughput gains might ignore the cost of online AnS computation and mixed-precision KV storage.

A: We clarified that all reported throughput already includes online AnS computation and FP16 sliding-window KV (<0.1% of KV memory). We also emphasized that the 840K-context number is measured end-to-end with real GPU memory usage (nvidia-smi + profiler), not a theoretical extrapolation.

---

> ### Author Response · Authors · 2025-12-04
> **Recognition from Reviewers**
>
> We sincerely appreciate the reviewers’ positive recognition of our work. Their feedback highlights the novelty, thoroughness, and practical relevance of our proposed framework, and we are grateful for their constructive and encouraging evaluations. Below, we summarize the key aspects of the reviewers’ recognition.
>
> ### [Novelty & Significance]
> - Reviewer ekvD: “Using anchor tokens to safeguard quantization quality is a well-motivated idea… particularly important when we come to sub-bit quantization.”
> - Reviewer wueT: Highlights that the intuition that some tokens are more sensitive to quantization is sound and empirically validated.
> ### [Comprehensive Evaluation & Analysis]
> - Reviewer wueT: Notes that the study is comprehensive, with evaluations across benchmarks and model types and detailed analyses of token locality.
> - Reviewer ekvD: Acknowledges that AnTKV achieves state-of-the-art performance in ultra-low-bit settings.
> - Reviewer z2EB: Describes the dual-stage framework as a creative synthesis of vector quantization and token-importance discrimination.
> ### [System Performance & Practicality]
> - Reviewer ekvD: Praises the coherent integration with FlashAttention and the system-level engineering behind the custom GPU kernels.
> - Reviewer wueT: Recognizes that the method enables very long contexts (up to 840K tokens) on a single A100 GPU, addressing a key bottleneck in long-context LLM inference.

---

> ### Author Response · Authors · 2025-12-04
>
> We would like to respectfully bring the following to the AC’s attention.
> According to the automated assessment on https://iclr.pangram.com/reviews?submission_number=353
> , the review submitted by reviewer z2EB is flagged as “fully AI-generated” by the platform’s detection system.
>
> We fully acknowledge that such systems may not be perfectly accurate, and we do not claim that the reviewer actually used AI. However, given the importance of maintaining the integrity and fairness of the review process, we kindly ask the AC to verify whether this automated flagging requires any further attention or follow-up.
>
> We deeply appreciate the AC’s time and effort in overseeing the reviewing process.

---

### Note · Authors · 2026-01-26

I have read and agree with the venue's withdrawal policy on behalf of myself and my co-authors.

---

### Meta-Review · Area_Chair_y39i · 2025-12-11

**Summary:**

This paper studies ultra low bit KV-cache compression for LLMs, combining mixed-precision and vector quantization. It extends the attention sink work by introducing an anchor score that measures each token’s sensitivity to quantization. The method operates in two stages: offline learning of vector quantization centroids, and online selection of anchor tokens followed by quantization of less sensitive tokens. Overall, KV-cache compression is an important direction for efficient LLM, and the idea of using anchor tokens (a generalization of sink tokens) is interesting. However, reviewers have raised questions regarding the novelty of the contributions (many components overlap with prior methods) and the paper lacks ablation studies on the impact of several algorithmic choices.

I also read the paper carefully myself since there was no reviewer-author discussion this year and have some additional questions and suggestions:

The paper adopts a pre-rope strategy similar to Hooper et al. (2024). I would like more clarification on how applying rope on-the-fly at inference time increases latency, particularly for long sequences.

In the same vein, I am curious about the implications of Theorem 1 for quantization under rope. How exactly does RoPE interact with quantization error in the KV cache, and does the theorem provide any guidance?

Most of the current benchmarks are long-prompt or QA style datasets. In practice, these tasks tend to be relatively easier for KV-cache compression because responses are short and quantization error does not propagate very far. It would be interesting to include more challenging reasoning tasks such as math 500 using reasoning models.

There is a growing body of work on vector quantization based KV-cache compression. It would be valuable to include comparisons with recent methods such as: Palu: Compressing KV-Cache with Low-Rank Projection and CommVQ: Commutative Vector Quantization for KV Cache Compression.

Given that the method involves online token selection and vector quantization, it would be interesting to report latency overheads introduced by the algorithm.

In Equation (5), for the anchor score AnS(V), what does the index i range over? Is it summing over all tokens after token j (including j itself)? If so, earlier tokens might have more terms contributing to the sum. Does this bias them toward having larger anchor scores?

The writing in Lines 252 to 260 could be clearer. What exactly happens during the decoding stage? Do the authors simply apply slice attention?

**Reviewer Concerns:**

Several questions were clarified during the rebuttal period (e.g., the theoretical derivation and the comparison with alternative ways to compute the anchor token score). However, some concerns remain unresolved, such as the novelty of the contribution and the hardware challenges associated with mixed-precision KV caching.

**Reviewer Scores:**

Overall, I believe the paper could benefit a lot from the reviewer-author discussion, and reviewers may raise their scores. Nevertheless, because several key weaknesses remain unaddressed, the paper may still ultimately fall within the rejection range.

---

### Decision · Program_Chairs · 2026-01-26

Reject